# Bond engineering of molecular ferroelectrics renders soft and high-performance piezoelectric energy harvesting materials

Yuzhong Hu [1,2,10] ✉, Kaushik Parida [3,4,10], Hao Zhang[3], Xin Wang[5], Yongxin Li [6], Xinran Zhou[3], Samuel Alexander Morris[7], Weng Heng Liew[8], Haomin Wang[3], Tao Li[3], Feng Jiang[3], Mingmin Yang[2], Marin Alexe [2], Zehui Du[9], Chee Lip Gan[3], Kui Yao [8], Bin Xu [5], Pooi See Lee [3] ✉ & Hong Jin Fan [1] ✉

Piezoelectric materials convert mechanical stress to electrical energy and thus are widely used in energy harvesting and wearable devices. However, in the piezoelectric family, there are two pairs of properties that improving one of them will generally compromises the other, which limits their applications. The first pair is piezoelectric strain and voltage constant, and the second is piezoelectric performance and mechanical softness. Here, we report a molecular bond weakening strategy to mitigate these issues in organic-inorganic hybrid piezoelectrics. By introduction of large-size halide elements, the metal-halide bonds can be effectively weakened, leading to a softening effect on bond strength and reduction in polarization switching barrier. The obtained solid solution $C_6H_5N(CH_3)_3CdBr_2Cl_{0.75}I_{0.25}$ exhibits excellent piezoelectric constants ($d_{33} = 367$ pm/V, $g_{33} = 3595 \times 10^{-3}$ Vm/N), energy harvesting property (power density is 11 W/m$^2$), and superior mechanical softness (0.8 GPa), promising this hybrid as high-performance soft piezoelectrics.

Piezoelectrics are materials that exhibit no inversion symmetric crystal structure and thus enables the conversion between mechanical and electric energy. In addition to their applications as actuator, transducer and sensor[1,2], they are also proposed for energy harvester and new venues in soft electronics such as soft robots, electronic skins, and biomedical devices[3–5]. However, practical application of piezoelectric energy harvester is largely limited by low energy output performance[6], while soft electronics require properties of both high piezoelectricity and mechanical softness. These issues are practically challenging in conventional ferroelectrics since some of these properties are

mutually exclusive. The first is the conflict between piezoelectric performance and mechanical softness (low stiffness in linear elastic deformation region). In wearable devices, good softness is conducive to skin comfort and reduces interfacial delamination due to Young's modulus mismatch[5,7,8]. Specifically, under mechanical stress, the elastic modulus mismatch between hard piezoelectrics and soft electrode layer (or embed polymer matrix) induce different strain/deformation in these components, which results in crack and delamination at their interface and hence limits the device lifespan and performance stability. In addition, this elastic modulus mismatch between human body

[1]School of Physical and Mathematical Sciences, Nanyang Technological University, Singapore 637371, Singapore. [2]Department of Physics, The University of Warwick, Coventry CV4 7AL, UK. [3]School of Materials Science and Engineering, Nanyang Technological University, Singapore 639798, Singapore. [4]Department of Polymer and Process Engineering, Indian Institute of Technology, Roorkee, Uttarakhand 247667, India. [5]Jiangsu Key Laboratory of Thin Films, School of Physical Science and Technology, Soochow University, 1 Shizi Street, Suzhou 215006, China. [6]Division of Chemistry and Biological Chemistry, School of Physical and Mathematical Sciences, Nanyang Technological University, Singapore 637371, Singapore. [7]Facility for Analysis, Characterisation, Testing and Simulation (FACTS), Nanyang Technological University, Singapore 639798, Singapore. [8]Institute of Materials Research and Engineering, A*STAR (Agency for Science, Technology and Research), Singapore 138634, Singapore. [9]Temasek Laboratories, Nanyang Technological University, Singapore 637553, Singapore. [10]These authors contributed equally: Yuzhong Hu, Kaushik Parida. ✉e-mail: yuzhong.hu@warwick.ac.uk; pslee@ntu.edu.sg; fanhj@ntu.edu.sg

and rigid device leads to dislocation and stress at skin/device interface upon body movement, which results in wearing discomfort in smart electronics. However, despite the large number of piezoelectrics that have been explored since 1880, high softness and large piezoelectricity rarely come hand in hand. Inorganic piezoelectrics such as oxide perovskites have decent power output under mechanical stress and can possess $d_{33}$ above 200 pm/V[9], but the high mechanical stiffness (Young's modulus $c_{33}$ from tens to hundreds of GPa)[10] largely limits their application in soft and wearable electronics[11]. On the contrary, organic piezoelectrics such as Polyvinylidene fluoride (PVDF) have excellent elastic softness ($c_{33}$ about 2-3 GPa)[12] but inferior electro-mechanical performance ($d_{33}$ for example, are generally below 40 pm/V)[13].

Another pair is piezoelectric strain and voltage constants ($d_{ij}$ and $g_{ij}$). The piezoelectric coefficient defined as $d_{ij} = S_{ij}/E_{ij}$ describes the strain ($S_{ij}$) induced by electric filed ($E_{ij}$) and is one of the most important parameters defining of piezoelectrics. $g_{ij} = d_{ij}/\varepsilon_{ij}$ reflects the voltage ($V$) and energy ($U$) output of piezoelectrics under stress, which can be estimated by $V = g_{ij} \times stress \times thichness$ and $U = \frac{1}{2} \times d_{ij} \times g_{ij} \times stress^2 \times Volume$[6]. This makes $g_{ij}$ particularly important for harvester and sensor applications. The product of $d_{ij}$ and $g_{ij}$ is generally considered as the figure of merit (FOM) of piezoelectric energy harvesters[1,14,15]. However in general case, neither organics nor inorganic piezoelectric materials possesses both large $d_{33}$ and $g_{33}$, leading to their limited FOM and insufficient harvesting output, which is the main issue of piezoelectrics harvesters[6]. Specifically, in piezoelectric ceramics, the enhancement of $d_{33}$ is generally realized by morphotropic phase boundary (MPB) method, where the increase in piezoelectric coefficient is accompanied by dramatic rise in dielectric constant ($\varepsilon_r$)[16], leading to high $d_{33}$ but very limited $g_{33}$. For example, the $\varepsilon_r$, $g_{33}$ and $d_{33}$ for a typical lead zirconate titanate (PZT) are 2300, $20.2 \times 10^{-3}$ Vm/N and 410 pm/V, respectively, whereas organic piezoelectrics like PVDF have low $d_{33}$ (typically 33 pm/V) but much smaller dielectric constant ~13, resulting in a decent $g_{33}$ of $300 \times 10^{-3}$ Vm/N that is more than 10 times higher than general piezoelectric ceramics[17].

Organic-inorganic hybrid ferroelectrics (OIHF) are comprised of both organic and inorganic parts as building block. This mixed composition grants OIHF the potential of combined advantages of two features: one is the structure flexibility and low dielectric constant due to the organic composition[18]; the other is semiconductor properties such as photovoltaic effect owing to with the inorganic metal-halide polyhedra[19,20]. This low dielectric features makes OIHF especially promising for high-$g_{33}$ piezoelectrics[21]. In addition, the composition flexibility at the organic and halide sites provides an avenue for structural and bond engineering, leading to various outstanding electromechanical features. For instance, the MPB induces the large $d_{33} = 1540$ pC/N of $(TMFM)_{0.26}(TMCM)_{0.74}CdCl_3$[22] and the space confinement effect leads to giant shear strain/piezoelectricity of $C_6H_5N(CH_3)_3CdBr_xCl_{3-x}$ solid solution[23]. Here, we report the OIHF $C_6H_5N(CH_3)_3CdBr_2Cl_{1-x}I_x$, in which we have realized coexistence of the above-mentioned two pairs of multi-exclusive properties. The iodide substitution in this hybrid effectively weakens the Cd-X1 chemical bond, leading to soft 1D inorganic chain and largely flattened polarization switching barrier. This results in excellent mechanical softness and high piezoelectricity. Under similar compressive stress, the crystal shows a piezoelectric energy harvester FOM = $1.22 \times 10^{-9}$ m$^2$/N and output power density of 11 W/m$^2$ that are about two orders of magnitude higher than those of PVDF and PZT based harvester, and simultaneously its softness ($c_{33}$ = 800 MPa and compressive strain up to 15.7%) is superior to the mainstream soft piezoelectric materials such as PVDF ($c_{33}$ = 2 GPa) and other hybrid systems ($c_{33}$ ~ few tens of GPa)[24]. Owing to the features of high transparency, lightweight and facile synthesis, this solid solution crystal and the associated molecule engineering strategy may inspire judicious material designs for new functional piezoelectrics and significantly boost its applications in soft electronics.

## Result and discussion

X-ray diffraction (XRD), thermogravimetric analysis (TGA) and differential scanning calorimetry (DSC) measurements were first conducted to study the crystal structure, thermal stability and phase transition properties of this OIHF, respectively. As indicated by powder XRD and refinement results (Supplementary Fig. 1 and Table 1), the lattice shows monotonous expansion with increasing iodide ratio, while a $Cc$ to $Ama2$ transition occurs around $x = 0.25$. Both these two space groups correspond to ferroelectric structures, yet the $mm2Fm$ transition suggests the induction of symmetry along the [001] crystal direction[25], which can be reflected by the merge of (11-1) and (111) XRD peaks (Supplementary Fig. 1b). TGA measurement conducted on crystal sample (Supplementary Fig. 2a) indicates that the material is stable up to around 200 °C. According to DSC analysis (Supplementary Fig. 2b), with $x$ increasing from 0 to 0.25, the transition temperature ($T_p$) decreases from 135 to 58 °C, while no phase change occurs for crystals with $x > 0.25$ until their decomposition around 200 °C.

Single-crystal XRD studies were then conducted to investigate the structure of this solid solution. The crystallography in low temperature phase (LTP) for $0 \le x \le 0.25$ adopts a one-dimensional (1D) hybrid structure with $Cc$ space group, where the inorganic chain consists of edge-sharing CdX$_5^-$ hexahedra and organic cations $C_6H_5N(CH_3)_3^+$ filling the space encapsulated by the metal-halide backbone (Fig. 1a). The organic and inorganic moieties are connected by electrostatic force and weak hydrogen bonds between halide anions and methyl group. By regarding Cd and N sites as the centers of negative and positive charges, respectively, spontaneous polarization can be identified along the [100] and [001] direction (Supplementary Fig. 3). However, the polarization along the [100 direction is non-switchable as evidenced by the temperature dependent dielectric behaviors (Supplementary Fig. 4) and the persistence of spontaneous polarization along the [100] direction in the high temperature phase (HTP, Supplementary Fig. 3b). High temperature and increasing iodide concentration above $x = 0.25$ trigger similar $Cc$ to $Ama2$ structure transition (Fig. 1b, c, respectively). During this phase transition, the longest metal-halide bond Cd-X1 breaks due to thermal energy (increasing halide size at X1 site in the situation of composition induced structure transition), and transfers metal-halide hexahedron into CdX$_4^-$ tetrahedron with a mirror symmetry along the 1D chain direction. The change in inorganic backbone induces two degenerated energy-minimum positions for organics cations with mirror symmetry regarding the [001] plane, which results in an order-disorder transition in cation and the loss of net polarization along the [001] direction.

**Table 1 | Structure information of this OIHF at room temperature**

| x | 0 | 0.1 | 0.2 | 0.25 | 0.27 | 0.5 | 0.75 | 1 |
|---|---|---|---|---|---|---|---|---|
| a | 12.9449(6) | 12.9449(6) | 12.9701(5) | 12.9681(4) | 12.9779(8) | 13.0376(4) | 13.1404(4) | 13.1393(3) |
| b | 14.7164(8) | 14.7269(5) | 14.7418(5) | 14.7417(5) | 14.7582(9) | 14.8133(4) | 14.9009(4) | 14.8910(4) |
| c | 7.3849(4) | 7.3885(2) | 7.4193(2) | 7.4176(2) | 7.6580(4) | 7.7154(2) | 7.8074(2) | 8.8140(2) |
| β | 95.211(4) | 95.173(3) | 95.000(3) | 95.000(3) | - | - | - | - |
| Space group | Cc | Cc | Cc | Cc | Ama2 | Ama2 | Ama2 | Ama2 |

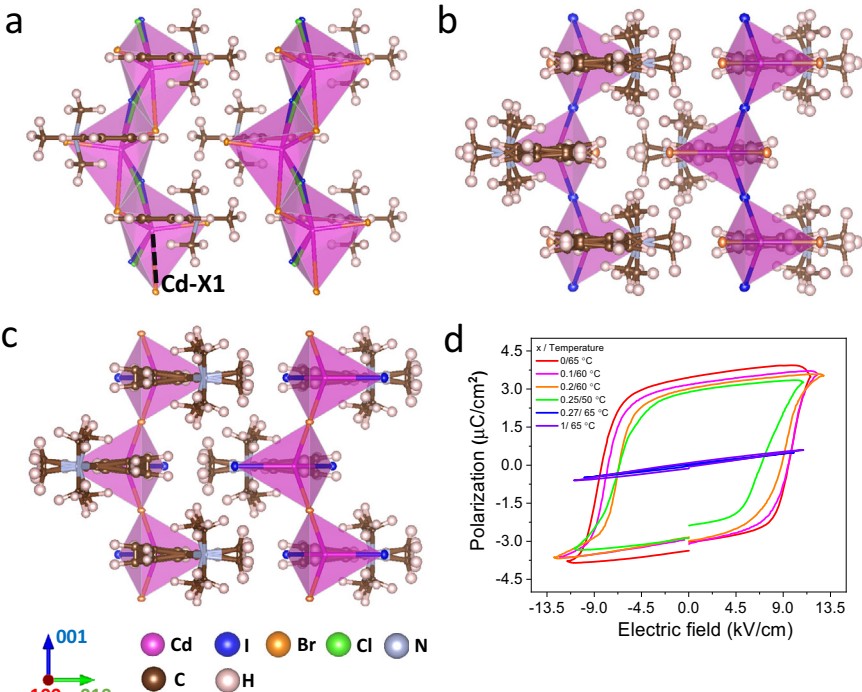

**Fig. 1 | Crystal structures and ferroelectric properties of $C_6H_5N(CH_3)_3CdBr_2Cl_{1-x}I_x$.** **a–c** Perspective of crystallographic structures of $x = 0.25$ crystal in low (**a**) and high (**b**) temperature phase and structure of $x = 1$ composition (**c**). **d** *PE* diagram of this solid solution with selected halide compositions, electric field along the [001] direction with frequency of 1 Hz.

The space-confinement structure in $x \leq 0.25$ crystals grants the lattice with large ferroelastic strain output upon polarization switch[23]. This is evidenced by the temperature triggered *mm2Fm* transition and the persisting [100] spontaneous polarization at HTP. Specifically, during the polarization switch under the [001] electric field, the coupling effect between the large-size organic moiety and limited space encapsulated by inorganic framework disables 180° polarization reversal (Supplementary Fig. 5a) and results in a $2\beta$ ferroelastic rotation[23], where $\beta$ is the monoclinic angle of the lattice. To confirm the ferroelectricity and non-180° ferroelastic switch of this solid solution, polarization-electric field (PE) and strain-electric field (SE) measurements were carried out by applying the electric field along the [001] direction at a certain minimum temperature to facilitate polarization switch[18]. As illustrated in Fig. 1d, with x increasing from 0 to 0.25, the remnant polarization shows slight decrease from 3.41 to 2.76 $\mu C/cm^2$, while the coercivity ($E_c$) decreases from 8.91 to 6.75 kV/cm, and the minimum temperature also drops from 65 to 52 °C. This indicates the iodide substitution can effectively reduce the ferroelastic switching barrier. To demonstrate the resultant ferroelastic strain upon this dipole rotation, the lateral displacement of the crystal upon polarization switching was measured by using a photonic sensor (Supplementary Fig. 5b, c). With [001] regarded as the direction **3** (Voigt notion), the produced ferroelastic strain in lateral direction **5** should be $S_5 = \tan\alpha \approx d/T$, where $\alpha$, $d$ and $T$ are the rotation angle, lateral displacement and sample thickness, respective[26]. As illustrated in Supplementary Fig. 5c, the $S_5$ obtained in the $x = 0.25$ crystal reaches 15.9% (corresponding to $\alpha = 9.09°$). This is consistent with the $2\beta = 9.94°$ obtained from crystallography measurements (Supplementary Table 1). With x increasing above 0.25, the crystal shows no polarization hysteresis or ferroelastic feature due to the loss of spontaneous polarization along [001] direction.

The robust ferroelastic switching and iodide substitution induced softening effect endow this OIHF with superior piezoelectricity. The direct piezoelectric effect was first studied by applying dynamic stress on Au-Ti/single crystal ([001] oriented)/Ti-Au device and measuring the generated current on a homemade tester[27]. As illustrated in Fig. 2a, under sinusoidal stress $\sigma(t)$, current $J(t)$ with same frequency $f$ was generated and piezoelectric coefficient can be obtained by $d_{33} = J_0/2\pi f \sigma_0$, where $J_O$ and $\sigma_O$ are the magnitude of current density and stress curves, respectively[27]. The $d_{33}$ shows stress and frequency dependent behavior (see Supplementary Fig. 6), which is similar to those of piezoelectric oxides with robust domain wall motion[28]. With increasing iodide ratio, a large $d_{33}$ of 324 pC/N is obtained in the $x = 0.25$ crystal under 2 MPa/1 Hz stress. As for the converse piezoelectricity, shear piezoelectric constant ($d_{35}$) was measured on pre-poled single crystals with the same experiment setup as shear strain measurements (Supplementary Fig. 5b). The SE curves obtained under unipolar electric field show hysteresis effect and increase in magnitude in crystals with higher iodide ratio (Supplementary Fig. 5d). This nonlinearity is attribute to the contribution of non-180° ferroelastic switch and the general low switching speed of hybrid ferroelectrics[18]. The 'large-signal' piezoelectric coefficient $d'_{35} = S_5/E_3$ is approximated by $S_{max}/E_{max}$, that is, the slope of SE curves (see Supplementary Text 2 for detail discussion)[29,30]. Similar frequency and electric field dependences were observed as in direct piezoelectricity measurements (Supplementary Fig. 5e). Under 1 Hz 6.5 kV/cm electric field, $d_{35}$ of the $x = 0.25$ crystal reaches 3100 pm/V.

Consider this coexisting $d_{35}$ contribution, the $d_{33}$ of this solid solution was then measured using a laser scanning vibrometer and mechanical pinning (Fig. 2b)[31]. With sinusoidal electric field applied in the [001] direction, the $d_{33}$ contribution can be extracted from the displacement magnitude diagram of the top surface[31,32]. Specifically, the single crystal was hanged in air by two thin probes pinning at each [010] surface center to ensure quasi-free boundary condition. In this case and for piezoelectrics with $Cc$ space group, both $d_{35}$ and $d_{33}$ will contribute to the displacement (D) of top surface (see Supplementary Text 2 for detail discussion). As shown in the simulation results on an example piezoelectrics $LiH_3(SeO_3)_2$ (with the same $Cc$ space group), the combination effect of $d_{33}$ and $d_{35}$ will lead to a valley-shape displacement magnitude diagram with zero-D line deviating from **X** = 0

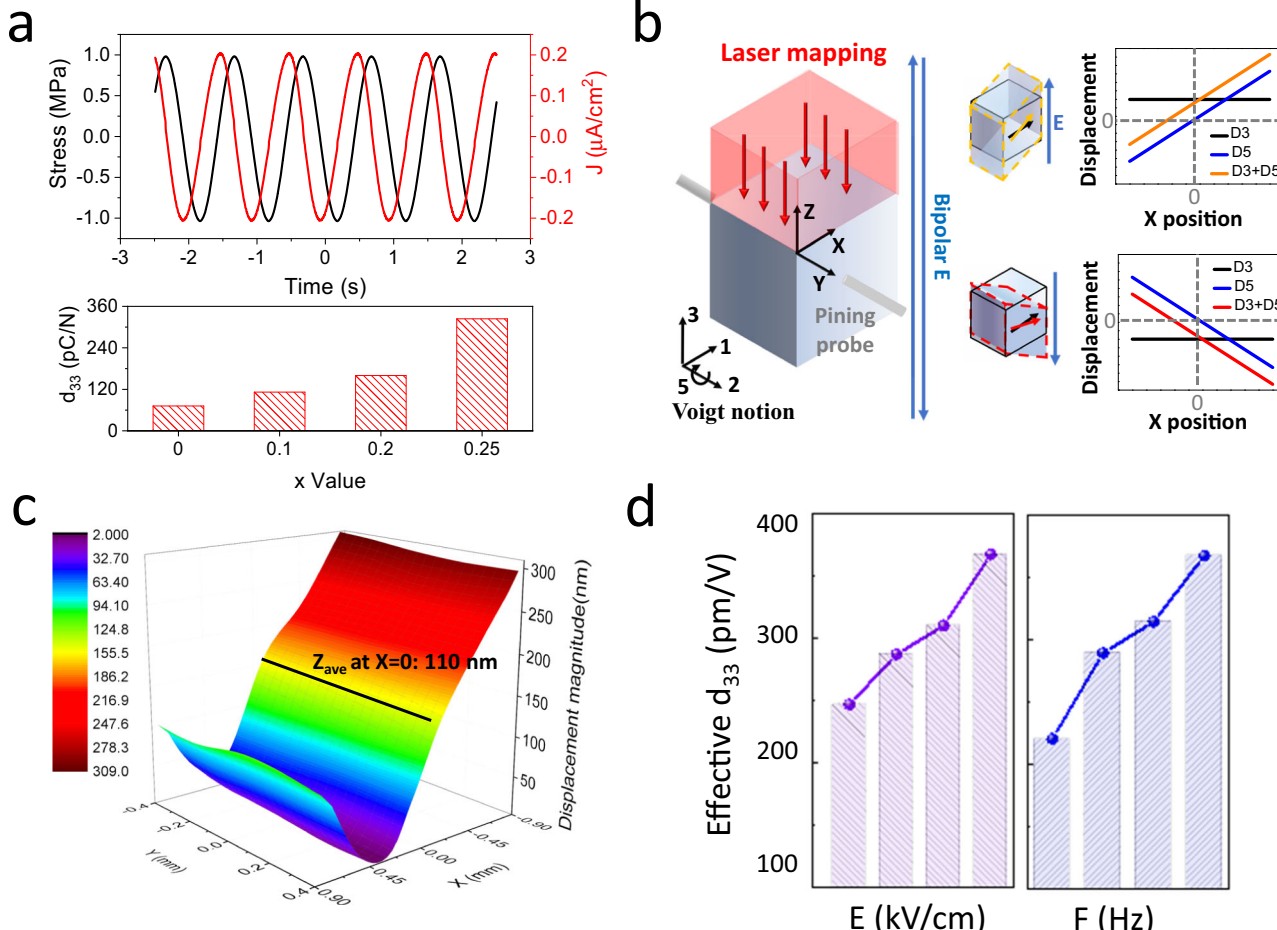

**Fig. 2 | Piezoelectric properties. a** Stress (1 Hz, 2 MPa) and generated current waveforms of $x = 0.25$ crystal (upper) and the $d_{33}$ of selected compositions obtained under the same stress (lower). **b** Experiment setup of converse $d_{33}$ measurement (left), simulated $LiH_3(SeO_3)_2$ (piezoelectric materials with the same $Cc$ space group) crystal deformation (middle) and corresponding displacements of top surface under electric field (right). $D_3$ and $D_5$ are the displacements caused by $d_{33}$ and $d_{35}$, respectively. The **x**, **y** and **z** correspond to (100), (010), (001) crystal orientations, respectively. **c** Displacement magnitude diagram obtained on $x = 0.25$ crystal under **E** = 2.4 kV/cm and $F = 1$ Hz bipolar electric field. **d** Electric field (left) and frequency (right) dependent $d_{33}$ of $x = 0.25$ crystal measured by converse piezoelectric effect.

line, where the normal ($S_3$) and shear strain ($S_5$) offset each other (see Fig. 2b and Supplementary Fig. 7b–e, X is the coordinate axis parallel to [100] orientation). The $d_{33}$ contribution can be extracted at **X** = 0 line, where $D$ is not affected by shear piezoelectricity contribution. As indicated in Fig. 2c, d, the average displacements at **X** = 0 position of $x = 0.25$ crystal reaches 110 nm under 1 Hz 2.4 kV/cm electric field, corresponding to a large $d_{33}$ of 367 pm/V, while its frequency and electric field dependencies show similar Rayleigh-like behavior. The consistent shape between our displacement magnitude diagram and the simulated results of $LiH_3(SeO_3)_2$ (Supplementary Fig. 7e) indicates the good reliability of our measurements.

Evidently, based on the above direct and converse piezoelectric measurements results, the piezoelectricity of this OIHF has obvious dependence on the magnitude and frequency of external stimulus, indicating the Rayleigh-like behavior and contribution from non-180° ferroelastic domain motion[33]. The great enhancement of piezo-electricity with iodide substitution is due to the induced lattice soft-ening effect, which can be reflected by the decrease in coercivity and lower temperature required for polarization switching in PE mea-surements. This barrier flattening effect is also evidenced by the decreasing barrier along the minimum energy path associated to polarization switch calculated based on density function theory (DFT, see Fig. 3a). Similar to the case in ferroelectric relaxors such as PZT and lead magnesium niobate-lead titanate (PMN-PT), this softening effect increases both the intrinsic piezoelectricity and extrinsic

electromechanical contribution by activating ferroelastic domain motion[28,34]. As illustrated in Fig. 3b, the $d_{33}$ of this OIHF is higher than most of ferroelectric polymers, composites materials and comparable to mainstream piezoelectric ceramics like PZT. More importantly, with its molecular ferroelectrics composition which generally corresponds to low dielectric constant[35], this OIHF exhibits a dielectric constant around 9, producing an ultrahigh $g_{33} = 3595 \times 10^{-3}$ VmN$^{-1}$ that is one to two orders of magnitude higher than most of organic and inorganic piezoelectric systems. The coexistence of high piezoelectric strain and voltage coefficient will endow this hybrid ferroelectrics with great potential in harvester applications[1,14].

In addition, the iodide substitution in this OIHF effectively redu-ces the strength of metal-halide bonds, leading to superior mechanical softness. To measure the elastic property, micropillars were first machined from the top surface of (001) single crystals using Focused Ion Beam (FIB) milling and then compressed by a nanoindenter (refer to Supplementary Fig. 8). As shown in Fig. 4a, with halide composition changes from pure Cl to $x = 0.25$, the Young's modulus lowers from 2.87 GPa to 800 MPa, while the fracture strain increases monotonously from 4.6% to 15.7%. This elastic softness of $x = 0.25$ crystal is superior to all piezoelectric ceramics, general metals, most piezoelectric organics and comparable to hard biological systems (Fig. 4b). The mechanical strength test was repeated on three $x = 0.25$ micropillars and standard Pt sample (Supplementary Fig. 8b, c). The reasonable data variation for $x = 0.25$ crystal and good consistency to previous reports on Pt

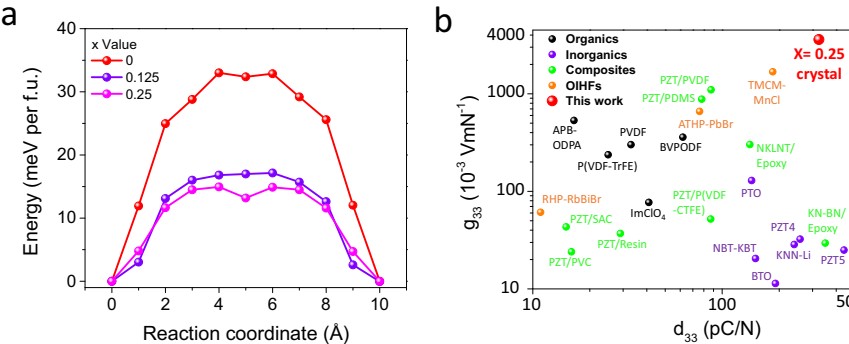

**Fig. 3 | Switching barrier simulation and piezoelectricity comparison.**
**a** Minimum energy paths calculated using DFT method. **b** $g_{33}$ and $d_{33}$ comparison of different materials (see Supplementary Table 2 for reference and details). For clarity, data for PMN-Pt ($d_{33}$ up to 2000 pC/N) and $(TMFM)_{0.26}(TMCM)_{0.74}CdCl_3$ ($d_{33}$ = 1540 pC/N), which are generally regarded as the piezoelectrics with the highest $d_{33}$ in inorganic and organic system, respectively, is not included in **b**.

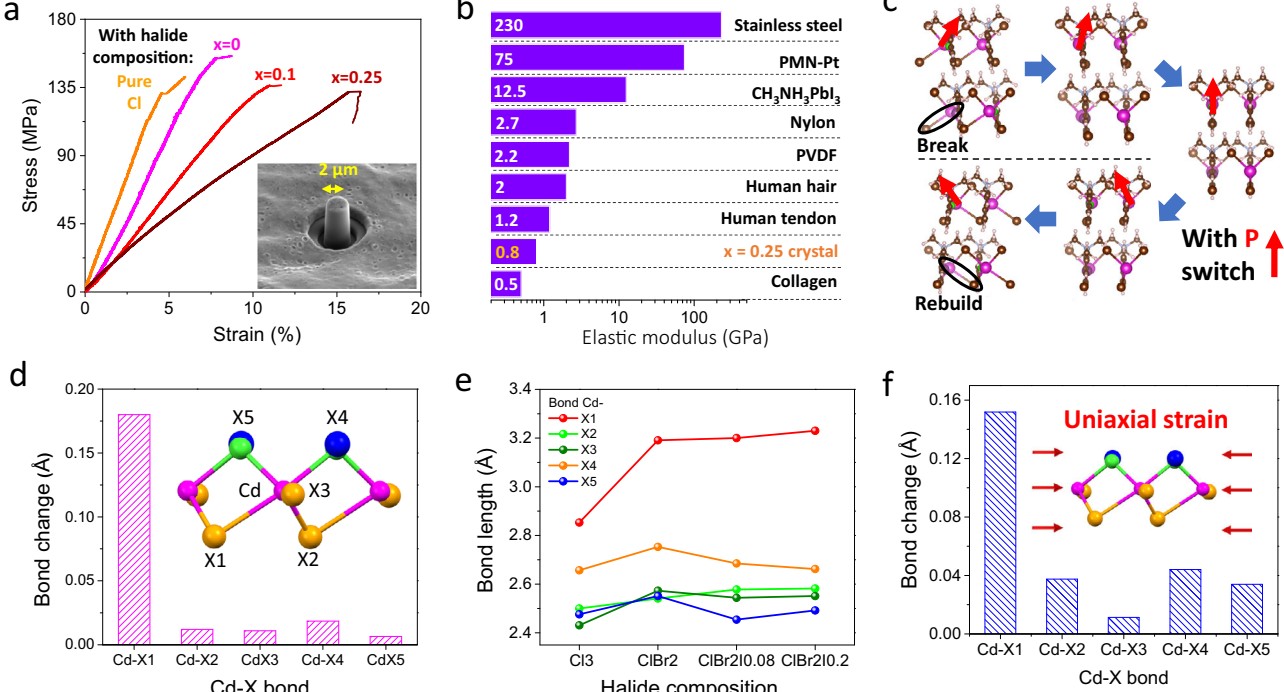

**Fig. 4 | Mechanical properties and mechanism. a** Strain-stress curves of $C_6H_5N(CH_3)_3CdX$ with different halide compositions. Inset: scanning electron microscope (SEM) image of a representative pillar for test. **b** Softness comparison of different materials and biological tissues. **c** Breaking and rebuilding of Cd-X1 bond upon polarization switch. **d** Cd-X bond length change at 90 °C of $x = 0.1$ lattice with respect to its room temperature structure. **e** Composition dependent bond length of different Cd-X bonds obtained by single XRD measurements. **f** DFT calculated bond length changes under 6% [001]-direction strain in $x = 0.25$ lattice.

sample[36,37] indicate the reliability of this mechanical measurement. To the best of our knowledge, the Young's modulus of $C_6H_5N(CH_3)_3CdBr_2Cl_{0.75}I_{0.25}$ shall be the lowest among all organic-inorganic hybrid systems[24]. In an organic-inorganic system, the mechanical properties are mainly governed by its internal structure, hydrogen and metal-halide bonds[24,38,39]. A weaker metal-halide bond will result in elastic softness[38,40]. Thus, in OIHF with 1D structure and weak CH-X hydrogen bond[41], the strength of metal-halide bond shall play the major role in its mechanical properties. In particular, the Cd-X1 bond, which is also the longest and weakest Cd-X bond, has a dominating influence on the mechanical property of the crystal. This can be reflected from the polarization switch process (Fig. 4c), where this weak bond is broken and rebuilt under applied electric field, leading to a large 9.94° (15.9%) ferroelastic switch (deformation) in the lattice. This special behavior indicates the excellent plasticity of this metal-

halide bond, which has been rarely observed in perovskite oxides as well as piezoelectric polymers.

As the in-situ measurement of the crystal structure under mechanical stress is extremely challenging, here we measure the temperature dependent crystal structure (with composition of $x = 0.1$) to verify the low stiffness of Cd-X1 and its crucial role in the lattice expansion of this OIHF. As shown in Fig. 4d, with temperature increased from 25 to 90 °C (below phase transition), the Cd-X1 bond shows a large expansion to 0.18 Å, while the rest Cd-X bonds have negligible changes (around 0.01 Å). This demonstrates the low energy required for bond length change and the dominant role of Cd-X1 bond in the expansion of metal-halide chain, which changes from 7.39 to 7.48 Å in 1D direction (Supplementary Fig. 9). In addition, the important role of Cd-X1 bond can also be reflected by the consistent relationship between the length change of Cd-X bonds and the mechanical

properties of this OIHF with different iodide compositions. As shown in Fig. 4e, with the halide site substituted by heavier element, the Cd-X1 shows large and monotonous expansion in length which corresponds to a decrease in bond strength, while the change on other metal-halide bonds are much smaller and show no evident tendency. The weakening effect on Cd-X1 bond following the increase of iodide ratio is also confirmed by the decreasing of integrated crystal orbital Hamilton populations up to the Fermi level (Supplementary Fig. 10). Finally, according to DFT calculation, the Cd-X1 bond illustrates a 0.152 Å decrease in bond length under 6% [001]-direction strain (Fig. 4f). This is significantly larger than those of other Cd-X bonds, corroborating the excellent elasticity and the crucial role of Cd-X1 in the mechanical softness of this OIHF.

The excellent piezoelectric and mechanical properties as well as optical transparency (Fig. 5a) make this OIHF crystal promising as energy harvester in future soft and wearable devices. However, up to date, explored OIHFs with decent piezoelectric response are in 1D structure and their $d_{33}$ direction are along the 1D inorganic chains[18,23]. This makes it very difficult to conduct precise energy harvesting measurement on bulk sample since the synthesis of 1D crystal with cross-section that is large enough to withstand compressive force requires impracticable long time (Supplementary Fig. 11). To mitigate this issue, we employed a special confinement method to synthesize single crystal wafer with large cross section area perpendicular to $d_{33}$ direction. A seed crystal is first placed with its [001] direction perpendicular to bottom glass slide. A spacer between the top and bottom slides determinates the thickness of the wafer. Herein, poly-dimethylsiloxane (PDMS) spacer was used and weight was added on the top slide to achieve tunable thicknesses (Fig. 5b and Supplementary Fig. 11c). With solution evaporation, single crystal wafer with area up to 1 cm² can be obtained with customized shapes such as Christmas tree, gear and bear (Supplementary Fig. 11d, e).

To measure the signal originating from piezoelectricity and minimize contribution from electrostatic or triboelectric effects, the harvesting property of this OIHF was evaluated from the single crystal wafer by sandwiching it between silver paste under compressive force. The voltage output shows monotonous increase with x going up to 0.25 (Supplementary Fig. 12a). A giant voltage output of 210 V and current density of 64 μA/cm² (corresponding to maximum power density of voltage × current density = 134 W/m²) were obtained on x = 0.25 single crystal, which is 10–20 times larger than the voltages of poly(vinylidene fluoride-trifluoroethylene) (P(VDF-TrFE)) film and PZT crystals (Fig. 6a) under same stress (3 MPa). The highest power density of 11 W/m² was obtained under a load impedance of 10⁶ Ω (Fig. 6b), which is orders of magnitude higher than those of other piezoelectric devices (Fig. 6d). Furthermore, the harvesting voltage output can be further improved by cascading several crystals (Supplementary

Fig. 12f), indicating the great potential to further enhance the performance.

The outstanding harvesting performance should be attributed to the coexistence of high $d_{33}$ and $g_{33}$. This leads to a giant FOM of $1.22 \times 10^{-9}$ m²/N that is around two to three orders of magnitude higher than conventional piezoelectrics such as PZT and PVDF (Fig. 6c). The comparison in Fig. 6d also shows clearly this OIHF crystal has a superior compatibility of softness and high performance among mainstream piezoelectric systems. In addition, this OIHF is transparent in the Ultraviolet−visible (UV-Vis) range (Fig. 5a), which is highly desirable in various piezoelectric applications including wearable devices, haptic actuator and photoacoustic transducer[42]. For example, in electronic skin, transparent material not only enable the integration of other wearable optoelectronics but also provide natural view on skin during daily activity[8,43]. Especially, the coexistence of high piezoelectric coefficient and optical transparency is rarely found in general piezoelectrics since high-performance piezoelectric ceramics always contain a large density of light-scattering domain walls[42]. Finally, a proof-of-concept wearable harvester device was demonstrated by sandwiching an array of x = 0.25 crystals within indium tin oxide and 'Very High Bond' (VHB) tape (Supplementary Fig. 13a). This transparent device exhibits harvesting voltages ranging from 230 to 260 V under mechanical bending up to 60° (Supplementary Fig. 13b−f).

In summary, we have demonstrated a molecular bond-weakening approach to realize a hybrid piezoelectric material in which excellent piezoelectric performance and mechanical softness capable to coexist. The result underpins the promise of piezoelectrics for applications in soft electronics and high-performance energy harvesting. Realization of the dual functions in this study highlights the important position of hybrids in the ferroelectrics family, which is promising to bridge the long-last gap between soft piezoelectric biosystem/polymer and conventional high-performance piezoelectric ceramics.

## Methods

### Synthesis of $C_6H_5N(CH_3)_3CdBr_2Cl_{1-x}I_x$ ($0 \leq x \leq 1$) single crystals

All chemicals were purchased from Sigma-Aldrich and Tokyo Chemical Industry and used without purification. $C_6H_5N(CH_3)_3CdBr_2Cl_{1-x}I_x$ single crystals were synthesized by a slow evaporation method. Specifically, $C_6H_5N(CH_3)_3$-Cl, $C_6H_5N(CH_3)_3$-I, and $CdBr_2 \cdot 4H_2O$ with stoichiometric ratios were dissolved in an acetonitrile (ACN) and deionized (DI) water mixed solution with ratio of 9:1. The solutions were then filtered by 0.45 μm filter after stirring for 6 hours. Prism shape crystals with centimeter length and few-millimeter width can be obtained upon weeks of slow evaporation. $C_6H_5N(CH_3)_3CdBr_2Cl_{1-x}I_x$ with x of 0, 0.1, 0.2, 0.25, 0.3, 0.4, 0.6, 0.75, 0.9 and 1 were synthesized by precursors with I molar ratio of 0%, 6%, 9%, 12%, 14%, 20%, 23%, 27%, 30% and 33%, respectively. Solution needs to be kept refreshed during evaporation

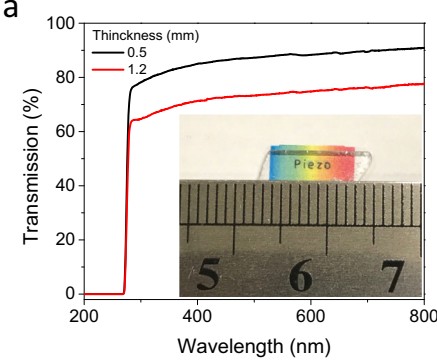
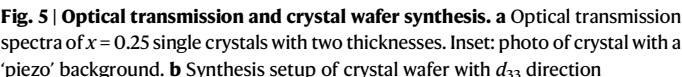
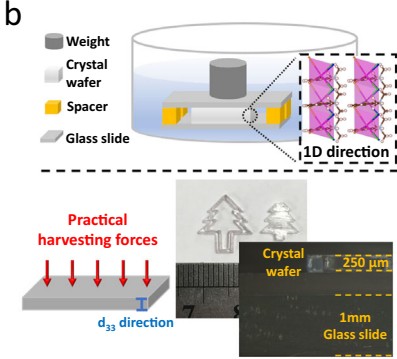

**Fig. 5 | Optical transmission and crystal wafer synthesis. a** Optical transmission spectra of x = 0.25 single crystals with two thicknesses. Inset: photo of crystal with a 'piezo' background. **b** Synthesis setup of crystal wafer with $d_{33}$ direction perpendicular to large-area plane. Bottom photos show crystals with a Christmas-tree shape and thickness of 250 μm.

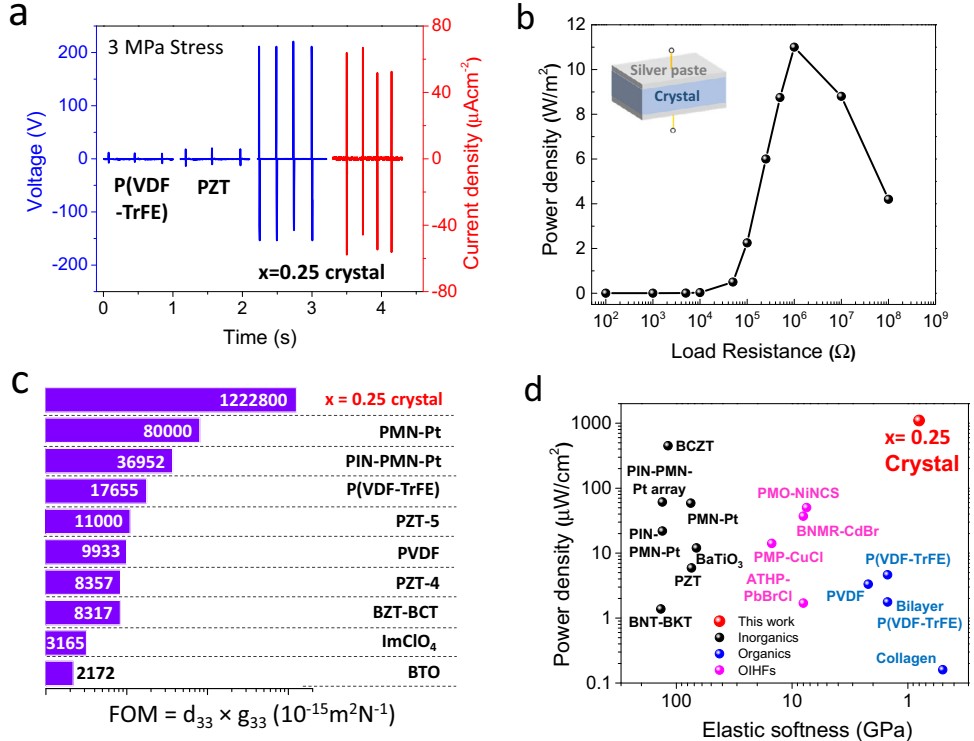

**Fig. 6 | Piezoelectric harvesting properties of Single crystal wafer. a** Voltage and current outputs compared with measured PZT and P(VDF-TrFE) sample with similar dimension (5*5*2 mm) and applied stress. **b** Power density under various impedance loading of $x = 0.25$ crystal. **c** Comparison of piezoelectric energy harvester FOM among different piezoelectric materials. **d** Comparison of power density and material softness among different materials (see Supplementary Table 4 for references and details).

as the halide composition is highly sensitive to iodide ion concentration. Iodide ratios were calculated by H, C, and N mass fraction that was measured by a PerkinElmer model 2400 Series II elemental analyzer.

## Synthesis of large single crystal wafers with preferable orientation

As show in Fig. 5b, a small prism seed crystal was first attached to the edge of polydimethylsiloxane (PDMS) film with [001] direction perpendicular to glass slide surface and then transferred to the space confined by two parallel slides separated by spacers. The spacer material can be such as glass slide, metal foil and PDMS. Saturated solution with corresponding precursor was then carefully added into the vial and left for evaporation at room temperature. Single crystal wafer up to centimeter-level diameter and thickness equal to the spacer can be obtained after evaporation. Alternatively, with PDMS films as spacer, weight can be placed on the top slide to control the thickness of the space hence the crystal. Wafers with customized shape can be obtained by placing seed crystal in plastic modes with desirable outline (patterns herein include bear, gear and Christmas tree).

## XRD measurements

Powder XRD measurements were carried out on a diffractometer (Panalytical Xpert) with Cu Kα radiation (40 kV and 30 mA, $\lambda = 1.540598$ Å,). Crystallography data were collected by Bruker APEX II diffractometer with Mo ($\lambda = 0.71073$ Å) as source material. The structures of crystals were solved by using Bruker SHELXTL Software Package and refined by Full-matrix least-squares on $F^2$. Non-hydrogen atoms were subjected to anisotropic refinement while hydrogen atoms were generated geometrically and allowed to ride in their respective parent atoms; they were assigned with appropriate isotropic thermal parameters and included in the structure factor calculations.

## DSC and TGA measurements

Phase transition properties of the crystals were investigated using differential scanning calorimetry on a TA INSTRUMENTS - Q10 instrument in heating and cooling cycle with a temperature rate of 10 K min⁻¹ under atmospheric pressure and nitrogen flow. TGA data were collected on a TA INSTRUMENTS – Q500 instrument under air with a heating rate of 10 K min⁻¹.

## Dielectric and ferroelectric measurements

Temperature-dependent dielectric measurements were conducted using a cryogenic micromanipulator probe station with a heating stage. Bulk crystals were carefully cut and polished in form of plates along crystallographic axes of [100], [010] and [001] with silver paste as top and bottom electrodes. Dielectric data were acquired from a commercial LCR meter (Agilent E4980A) with 1 V alternating current (AC) voltage. Ferroelectric measurements were performed on a commercial ferroelectric tester (Precision LC, Radiant technologies) equipped with a high-voltage amplifier (Precision 4 kV HVI, Radiant technologies).

## Piezoelectric and strain measurements

The direct piezoelectric effect was measured using a home-built Berlincourt piezometer on the [001] oriented crystal. Au/Ti (50/5 nm) electrodes were deposited on the polished top and bottom surfaces of the crystal. More details can be found from Supplementary text 1 and our previous work[27].

The setup for shear piezoelectricity measurement is illustrated in Supplementary Fig. 5b which is the same in our previous work[23]. Specifically, crystals were fixed on glass slide by double side tape with [100] plane as the basal plane. Crystals were carefully cleaved along [001] plane that were polished and coated with silver pastes as electrodes. An aluminum plate with double side tape was attached on the

top of crystal used as reflective mirror. The mirror's plane is perpendicular to the crystal top surface and laser beam. A commercial photonic sensor (MTI-2100 Fotonic) was used to measure the displacement of crystal top surface under [001]-direction electric field ($E_3$). All samples were poled before piezoelectric measurements. Shear strain ($S_5$) is obtained by $S_5 = \tan\alpha \cong d/T$, where $T$ is the height of crystal; $\alpha$ is the ferroelastic rotation angle; $d$ is the horizontal displacement. 'Large-signal' $d^*_{35}$ was then obtained by $d^*_{35} = S_5/E_3$.

$d_{33}$ from converse piezoelectric effect was measured by using a laser scanning vibrometer (PolyTech GmbH) with two mechanical pinning and a high voltage signal generator as voltage source (Fig. 2b). Specifically, bulk single crystals were first polished carefully into cuboid and coated with silver paste on top and bottom [001] planes as electrodes. The crystal was then fixed by pining probes at both [010] geometric centers. The crystal was placed with [001] plane perpendicular to laser beam. Thin gold wires were fixed on top & bottom within silver paste and linked to voltage source. With sinusoidal electric field applied, the displacement of the top surface was simultaneously measured by laser scanning vibrometer. $d_{33}$ is calculated by $2D_{X=0}/V$, where $D_{X=0}$ and $V$ are displacement at $X=0$ line (see Fig. 2c) and applied voltage, respectively. Please refer to Supplementary Text 2 for details.

### Nanoindentation measurements

Bulk single crystals were first oriented in [001] direction. The top surface was then carefully polished and machined into pillar by Focused Ion Beam milling system (FEI Nova 600i Nanolab). Low milling currents with amplitude ranging from 0.1 to 3 nA were applied in sequence to minimize the effects of Ga+ source on the pillars. The obtained pillars have a cylindrical shape with dimensions of ~ 1 μm × 5 μm (radius × height). The micro-compression tests were conducted on an in-situ nanoindenter (Hysitron PI 85) equipped with a flat diamond tip (2.5 μm in radius). Compressive loads were applied at a constant loading rate of 5 μN/s until samples failed. At the start of the test, the transducer of the nanoindenter was calibrated to ensure the accuracy of the measurements.

### Optical transmission measurement

Optical transmission spectra of bulk single crystals with 500 and 1200 μm thicknesses were measured by using a Shimadzu UV-2700 spectrometer in the wavelength range from 200 to 800 nm.

### Harvester measurements

[001] oriented single crystal wafer with cross section around 5 × 5 mm and thickness 1–2 mm are synthesized by growth confinement method were used for harvester measurements. Conductive silver slurry was used as top and bottom electrodes and was connected to the measuring instrument by copper wires. Scotch tape was used to fix the single crystal and isolate the electrode from the metal rod (where the force is exerted, see set up in Fig. S14a). The flexible piezoelectric energy harvester was fabricated by poled [001] oriented single crystals with ~2 × 2 × 1 mm dimensions between two VHB tapes. The weight and volume ratio of the OIHF crystal in the harvesting device are 12.1% and 4.6%, respectively. Indium tin oxide (ITO) coated Polyethylene terephthalate (PET) films were utilized as the electrodes. A mechanical shaker (Sinocera, Model JZK-20) was used to exert dynamic mechanical impact on the device and a Force gauge (Sinocera, Model CL-YD-303) was utilized to measure the magnitude of the force. An oscilloscope (Trektronix, MDO 3024, impedance = 10 MΩ) was used to measure the voltage output from the energy harvester and the output current was measured using a low noise current pre-amplifier (Stanford Research System, Model SR570, impedance = 4 Ω). The power output of the device was evaluated by measuring the voltage output across various load impedances. The energy generated is utilized to charge an external capacitor via a full wave bridge rectifier. All the energy harvesting performance were performed in ambient environmental conditions (room temperature ~ 25 °C; relative humidity ~65%).

### Computation

(PTMA)CdBr$_3$Cl$_{(1-x)}$I$_x$ crystals ($x=0$, 0.125, 0.25) were simulated with explicit doping in a supercell using density functional theory, with four formula units for x = 0 and 0.25, and eight formula units for x = 0.125. The calculations were carried out using the projected augmented wave (PAW) methods, as implemented in the Vienna Ab Initio Simulation Package (VASP)[44,45]. The electron interactions were described using the Perdew-Burke-Ernzerhof (PBE) exchange-correlation functionals[46]. We explicitly treated 12 valence electrons for Cd ($4d^{10}5s^2$), 4 for C ($2s^22p^2$), 5 for N ($2s^22p^3$), 1 for H ($1s^1$), 7 for Cl ($3s^23p^5$), 7 for Br ($4s^24p^5$) and 7 for I ($5s^25p^5$). The plane-wave energy cutoff was set to be 500 eV, and the Brillouin zone was sampled by 2 × 2 × 4 ($x=0$, 0.25) and 2 × 2 × 2 ($x=0.125$) Γ-centered $k$-point grids. The lattice constants and atomic positions were fully optimized until the Hellmann-Feynman forces were less than 0.002 eV/A. Van der Waals interactions[47] were included in all calculations. The minimum energy paths for polarization switching were calculated using the generalized solid-state nudged elastic band (GSSNEB) method[48]. Crystal orbital Hamilton populations (COHP) were computed using the Lobster package[49,50]. The elastic constants were calculated from the strain–stress relationship with an energy cutoff of 800 eV.

### Reporting summary

Further information on research design is available in the Nature Research Reporting Summary linked to this article.

## Data availability

The data supporting the findings of this study are available within the article and its Supplementary Information on Dryad.

## Code availability

The data supporting the findings of this study will be available upon request.

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

## Acknowledgements

The authors thank J.Y. and S.P.N., School of Materials Science and Engineering, Nanyang Technological University, for facility support in single crystal synthesis, and X.P.L. and D.L., School of Engineering, the University of Warwick, for useful discussions about mechanical measurements. B.X. acknowledges financial support from National Natural Science Foundation of China under Grant No. 12074277 and Natural Science Foundation of Jiangsu Province (BK20201404), the startup fund from Soochow University, and the support from Priority Academic Program Development (PAPD) of Jiangsu Higher Education Institutions. K.Y. acknowledges the research supports by A*STAR, under RIE2020 AME Individual Research Grant (IRG) (Grant No.: A20E5c0086).

## Author contributions

Y.H., K.P., P.S.L., and H.J.F. conceived the idea and designed the project. Y.H., X.R.Z., and F.J. performed the powder XRD

measurements. X.R.Z. fabricated plastic outlines for crystal synthesis with customized shapes. S.A.M. conducted the powder XRD analysis. Y.L. and S.A.M. performed the crystallography XRD characterizations and analysis. H.Z., Z.H.D., H.M.W., and C.L.G. conducted FIB cutting and nanoindentation measurements. B.X. and X.W. carried out the DFT calculations. K.Y., Y.H., and W.H.L. designed, and W.H.L. performed $d_{33}$ measurements by converse piezoelectric effect, and K.Y. and Y.H. analyzed piezoelectric properties. K.P. fabricated harvester devices and conducted relevant characterizations. Y.H. synthesized the crystal, prepared the devices, simulated the displacement magnitude diagrams in converse piezoelectric measurements, and carried out the ferroelectric, shear piezoelectric, dielectric, transmission spectrum, TGA, DSC, elemental analyzer measurements. T.L. assisted the shear piezoelectricity characterizations. M.A. assist single crystal wafer growth. Y.H. and Y.M. carried out direct $d_{33}$ measurements. Y.H., K.P., H.J.F., and P.S.L. wrote the manuscript with input from all the authors.

## Competing interests

The authors declare no competing interests.
