## [Peer Review File · Nature Communications]

Bond engineering of molecular ferroelectrics renders soft and high-performance piezoelectric energy harvesting materialsREVIEWER COMMENTS

Reviewer #1 (Remarks to the Author):

Very good paper on organic materials for sensing and harvesting. Well written and presented. I only have minor comments:

“Softness” - seems vague - do you mean low stiffness?

Fig 2b - not clear what is the x - axis? No units and not defined?

Why has a standard Berlincourt meter not been used to measure d_{33} ? This will be beneficial.

Fig 3 b - small text and even harder to read the RED text, especially with purple background

Figure 3c - what is polarisation direction? Good to indicate in all figures with crystal structure.

Figure 4b. - very small and difficult to see

Porous ferroelectric also provide a route to reduce permittivity the sensing/harvesting figures of merit - see “Porous ferroelectric materials for energy technologies: current status and future perspectives M Yan et al. Energy & Environmental Science 2021

Signed: Chris Bowen

Reviewer #2 (Remarks to the Author):

Please see the attached file for a detailed review. The manuscript by Hu, Lee, Fan and co-workers describes the piezoelectric and energy harvesting properties of a compositionally tuned organic-inorganic hybrid ferroelectric, $C_6H_5(Me)_2CdBr_2Cl_{1-x}I_x$. This solid solution exhibits very high piezoelectric and other electromechanical coupling values for its $x = 0.25$ composition. The reported values of d_{33} , g_{33} and piezoelectric energy harvesting FOM values are very high for this class of materials. Hence, I would be glad to recommend the acceptance of this material in Nature Communications. However, given that this material shows the highest electromechanical coefficient values, the authors must address the following aspects before it can be suitable for publication.

1. The authors have mentioned that this material exhibits non-180 ° switching of its electric polarization. They must perform PFM measurements to demonstrate this aspect of this material, in addition to the reported temperature-dependent P-E loop measurements.
2. It would also be helpful to compare the d_{33} obtained from the home-built vibrometer to those extracted from the PFM spectroscopy measurements.
3. The authors have shown that the material exhibits very high output voltages and currents with very high FOM values. Considering that the measurements are done on the single crystal embedded between the VHB tapes, the performance reported is for the area of the VHB tape under study than that of single-crystal material. Hence, providing the wt% and vol% of the OIHF in the employed tape area is better.

4. Owing to high output voltages and current characteristics of the proof-of-concept device based on the single crystals, a detailed analysis of the piezoelectric energy harvester studies is required.

(i) To establish the piezoelectric origin of the obtained output voltages, the measurements should be performed by reversing the electrical connections to the device.

(ii) To rule out any possibility of the triboelectric effects in the measurements, single electrode or similar triboelectric-based measurements should be performed.

(iii) The energy harvesting efficiency of the device(s) must be reported.

(iv) A comparison of this single crystal device to its device based on bulk material (or its composite) must be performed to provide its practical utility.

5. Several tables in the manuscript and the supporting information are insufficient.

(i) Fig3b and Fig4e do not include sufficient data for the best performing OIH ferro- and piezoelectric materials reported earlier.

(ii) Table S2 shows no data of g_{33} and d_{33} for OIH and their composites. There are several recent articles and reviews on these materials have appeared in the recent past.

(iii) The authors have not mentioned the TMCM cation and their family of hybrids in any of their tables, as they exhibit the highest d_{33} values for any OIH material published so far.

6. Though the obtained output characteristics (voltage, power density) of the device are high, the voltages stored in the capacitors are very low and, in fact much lower than those obtained from several hybrid bulk composite materials (with no directional behaviour).

Hence, the authors must provide the data from the bulk sample and compare them with the literature known examples derived from hybrid materials.

7. The stored energies and measured charges in the capacitors should also be provided.

8. Most of the Figures in the manuscript are cluttered with multiple images and long figure captions. They must be split. Also, figure captions should be elaborated a bit more as this reviewer could not make a smooth correlation of the figure captions with the related text in the main manuscript.

Reviewer #3 (Remarks to the Author):

In this manuscript, the authors incorporated iodide into an organic-inorganic hybrid ferroelectric $C_6H_5N(CH_3)_3CdBr_2Cl$, which weakens the metal-halide bonds. The obtained $C_6H_5N(CH_3)_3CdBr_2Cl_{1-x}I_x$ ferroelectric solid solution shows outstanding piezoelectric and energy harvesting performances with g_{33} and output power density up to 3595×10^{-3} Vm/N and 11 W/m², respectively, much higher than those of widely used piezoelectric PZT. This work sheds light on the exploration of high-performance soft piezoelectrics. I recommend publishing it on Nature Communications after addressing the following issues.

1. In the introduction part, the authors only discussed the g_{33} of inorganic PZT and organic PVDF. The works on the g_{33} of organic-inorganic hybrid ferroelectrics were not mentioned (such as J. Am. Chem. Soc. 2020, 142, 1077–1082).

2. The piezoelectric performance of ferroelectric solid solution $C_6H_5N(CH_3)_3CdBr_3xCl_3(1-x)$ was previously reported by the authors. They should mention this in the introduction part as well.

3. For the d_{33} , the authors are suggested to measure it by the Berlincourt method.

4. The Young's modulus of organic-inorganic $C_6H_5N(CH_3)_3CdBr_2Cl_{0.75}I_{0.25}$ is even lower

than that of pure organic PVDF. The authors should double check the Young's modulus of $C_6H_5N(CH_3)_3CdBr_2Cl_{0.75}I_{0.25}$. The theoretical value is suggested to be provided.

5. In figure S4, the change of dielectric constant around T_c is very small, which is not common for ferroelectric crystals. Please discuss the reason.

Response to Reviewers' Comments

Page 1 – 4: Responses to Reviewer #1

Page 5 – 22: Responses to Reviewer #2

Page 23 – 27: Responses to Reviewer #3

Page 28 – 30: Reference

Reviewer #1 (Remarks to the Author):

Very good paper on organic materials for sensing and harvesting. Well written and presented. I only have minor comments:

Response: We thank Prof. Bowen for his positive comments.

1. “Softness” - seems vague - do you mean low stiffness?

Response: Yes, softness by definition is the ability of matter to easily deform under external forces, so it is very similar to low stiffness in linear elastic deformation region.

Action taken: We have added on the page 3 the following text: “softness (low stiffness in linear elastic deformation region)” to make this clear.

2. (i) Fig 2b - not clear what is the x - axis? No units and not defined?

(ii) Fig 3 b - small text and even harder to read the RED text, especially with purple background

(iii) Figure 3c - what is polarisation direction? Good to indicate in all figures with crystal structure.

(iv) Figure 4b. - very small and difficult to see

Response: We thank the Reviewer for spotting these issues. We have changed accordingly.

Action taken:

(i). In the updated Fig. 2b, the x is the coordinate position on crystal top surface. The direction of x is the same as (100) crystal orientation.

(ii). We have changed the color and deleted some non-essential information to make the figure clearer (it is now Fig. 4b).

(iii). We have indicated the polarization direction in Fig. 4c.

(iv). We have split Fig 4 to new Fig 5 and 6 and enlarged all parts.

3. Why has a standard Berlincourt meter not been used to measure d_{33} ? This will be beneficial.

Response: We are sorry that we have only cited our previous work¹ rather than explaining the measurement in detail in our experiment method part. Our system is Berlincourt piezometer. We did not use a commercial one because most of them are measuring at constant stress amplitude and frequency, which is not very helpful for us to investigate the mechanism of direct piezoelectric effect of this novel OIHF system. Instead, we have developed a home-build system which is in principle identical with the original Berlincourt piezometer but with tuneable stress frequency and amplitude (see Fig R1a). The noise floor, accuracy and linearity were checked by several piezoelectric and non-piezoelectric systems. The system provides high accuracy and has enabled the successful characterisation of artificially induced piezoelectricity in Schottky junctions (see Nature 584, 377–381, 2020)¹.

Regarding the present OIHF system, as mentioned in the main text, the converse piezoelectricity of this particular solid solution shows an obvious dependency on the magnitude and frequency of applied electric field. This Rayleigh-like behaviour due to domain reorientation shall have a considerable contribution to the piezoelectric effect². Considering this piezoelectric feature, we employed the classic Berlincourt piezometer (with tuneable stress frequency and amplitude) to measure the d_{33} value based on direct piezoelectric effect.

Details about the Berlincourt piezometer method and related equations have been added to Supplementary Text 1 and data was added as new Fig. S6a (see also Fig R1 below). Here let's mention briefly the result. At constant outer electric field condition (which is zero), the d_{33} of the tested sample can be obtained from:

$$d_{33} = J_0 / 2\pi f T_0$$

In addition, we also measured the piezoelectric d_{33} values of PZT ceramic and PMN-0.3Pt crystal (Fig. R1b). The obtained d_{33} of 440 pC/N for PZT and 1033 pC/N for PMN-0.3PT are consistent with previous reports^{3,4}, indicating the good reliability of our measuring system.

Fig. R1| a, Set up of our Berlincourt piezometer. **b**, Generated current density of $x = 0.25$ crystal, PZT ceramics and PMN-0.3Pt crystals under 2MPa 1 Hz AC stress.

Action taken: We have added the PZT and PMN-Pt data and full explanation in revised Supporting information (See Fig. S6a and Supplementary Text 1).

4. Porous ferroelectric also provide a route to reduce permittivity the sensing/harvesting figures of merit - see “Porous ferroelectric materials for energy technologies: current status and future perspectives M Yan et al. Energy & Environmental Science 2021

Action taken: We thank Prof. Bowen for his advice. We have cited this work in the revised introduction part (ref.8).

Reviewer #2 (Remarks to the Author):

Please see the attached file for a detailed review. The manuscript by Hu, Lee, Fan and co-workers describes the piezoelectric and energy harvesting properties of a compositionally tuned organic-inorganic hybrid ferroelectric, $C_6H_5(Me)_2CdBr_2Cl_{1-x}I_x$. This solid solution exhibits very high piezoelectric and other electromechanical coupling values for its $x = 0.25$ composition. The reported values of d_{33} , g_{33} and piezoelectric energy harvesting FOM values are very high for this class of materials. Hence, I would be glad to recommend the acceptance of this material in Nature Communications. However, given that this material shows the highest electromechanical coefficient values, the authors must address the following aspects before it can be suitable for publication.

Response: We thank the reviewer for his/her positive comments.

1. The authors have mentioned that this material exhibits non-180° switching of its electric polarization. They must perform PFM measurements to demonstrate this aspect of this material, in addition to the reported temperature-dependent P-E loop measurements.

Response: We have not attempted to demonstrate the ferroelastic switch by conducting temperature-dependent PE loop measurement. The PE data in Fig. 1d is composition dependent ferroelectric hysteresis measurements rather than temperature dependent. Due to the different switching barriers, different temperatures are applied to facilitate the polarization switching, which is a common practice for OIHF^{5,6}.

To the best of our knowledge, PFM is not an appropriate or necessary method to demonstrate the non-180° polarization switching in this OIHF bulk crystal, which exhibits large strain during polarization reversal. While PFM is basically a surface technique and applies perfectly to thin films and nanoscale structures, it has been regarded not suitable for studying switching behavior of bulk samples. Generally, the electric field penetration depth is no larger than 2-3 times of the radius of tip-surface contact diameter, so the penetration is few tens of nm. This penetrate depth is not enough to switch the domains in bulk ferroelectric crystals in few hundreds micrometer thickness. In addition, the radial distribution of the electric field is highly non-uniform within the bulk sample (see Fig. R2a)^{7,8}. The dimension of our single crystals is in range of hundreds of micrometers with coercive field around 600 V/mm even at elevated temperatures. Hence, the large sample thickness, high voltage required, very limited E penetration depth and non-uniform electric field make the polarization switching very hard to be achieved by PFM tip.

Fig. R2 | The radial electric field (upper) and electric field magnitude-penetration depth diagram of PFM tip (lower), where r_c is the radius of tip-sample surface⁷.

Without the PFM measurements, we already have several evidences to the non-180° switching. First evidence is from crystallography structure. The most intrinsic evidence is the crystallography structures at high and low temperature phases (HTP and LTP). As mentioned in the main text, the unit cell goes through an $Ama2-Cc$ (Orthorhombic to monoclinic) phase transition with temperature decrease (see Fig. R3). From Aizu rule, symmetry breaking caused by phase transition shall govern the ferroelectric features at LTP¹⁰. Specifically in this crystal, mirror symmetry is induced in 1D chain direction. So, there shall be two equivalent ferroelectric states at LTP and the monoclinic angle at LTP ($\beta = 4.971^\circ$) will determine the magnitude of produced non-180° ferroelastic strain during polarization reversal¹¹.

Fig. R3 | Phase transition and crystal structure of $x = 0.25$ composition at HTP and LTP.

Secondly, this non- 180° ferroelastic switch has been proved directly by our photonic sensor (Fig. S5c). With electric field applied along (001) direction, this ferroelastic switch shall produce shear strain with magnitude of 2β (Fig. R4a) during polarization switch, corresponding to a displacement of $l = H \times \sin(2\beta)$ in horizontal direction, where l and H are crystal thickness and displacement, respectively (Fig. R4b). With this setup, PE and SE loop of the $x = 0.25$ crystal are measured as shown in Fig. R4c (from Fig. 1d) and Fig. R4d (same as Fig. S5), respectively. The obtained ferroelastic angle of 9.14° (15.9% strain) is in good consistence with the monoclinic angle ($2\beta = 9.9^\circ$) obtained by crystallography measurement. The optical photos switched to two states also demonstrate this ferroelastic switch (Fig. R4e and f). These direct and consistent measurement result provide solid proves to the existence of the non- 180° ferroelastic switch.

Fig. R4 | Measurements about the two ferroelastic states (**1** and **2**) of $x = 0.25$ composition. **a**, Two ferroelastic states at Cc phase. **b**, Experiment setup for shear displacement measurement. Sample is fixed on slide with crystal orientation corresponding to that of **a**. **c**, Polarization-electric field loop. **d**, Strain-electric field loop. **e** and **f**, Optical photos of the crystal at **1** and **2** ferroelastic state (switched by electric field). The scale bar is 0.2 mm. The small flake at the top of crystal is part of the reflective mirror as shown in **b**.

Thirdly, the d_{33} shows distinct dependence on the magnitude and frequency of external stimuli by both direct and converse piezoelectric effects. This Rayleigh-like behaviour is obviously different

from the piezoelectricity due to lattice change and supports its connection with non-180° domain reorientation^{2,12}.

2. It would also be helpful to compare the d_{33} obtained from the home-built vibrometer to those extracted from the PFM spectroscopy measurements.

Response: Similar to the Comment 1 above, we would like to say PFM is not suitable to conduct quantitative piezoelectric measurement on this bulk sample, especially for the particular crystals in this study which have considerable shear piezoelectric contribution. Our arguments are as follows.

Firstly, PFM has intrinsic limitations for bulk crystals. As confirmed by several researchers in the PFM field, including Alexei Gruverman, Dennis Meier, Gustau Catalan and one of co-authors, Marin Alexe, the inhomogeneous electric field, limited penetration depth, and surface mechanical clamping will largely influence the piezoelectric signal and make accurate d_{33} measurement impractical on bulk samples^{7,8,13}. For example, the non-uniform electric field will induce piezoelectric lattice expansion and non 180° domain reorientations at different scale varying the depth of crystal (because of different electric field), whereas an accurate piezoelectricity measurement requires electric field to be uniformed in $d_{33} = \text{strain}/\text{electric field}$ ¹⁴. The PFM technique is thus more suitable to be employed in thin films and nanoscale piezoelectric characterization where these factors can be largely neglected.

Secondly, the bottom fixed boundary condition in PFM measurement and the considerable shear piezoelectric effect of this solid solution will make the displacement signal highly position dependent, and consequently make d_{33} extraction very inaccurate. As mentioned above, PFM is not capable of applying a uniformed electric field with large penetration depth. As shown in Fig. R5a-c and the discussion in Supplementary Text 2, the vertical displacement of a point on top surface can vary from almost zero to hundreds of nanometers even in quasi-free boundary condition. Under bottom fixed boundary condition (the crystal bottom fixed on PFM holder), the crystal shall move in lateral direction and thus make the top surface displacement complicate (Fig. R5d). Accordingly, the top and side crystal surfaces will expand and tilt due to shear piezoelectric contribution (can refer to Fig. R5a and estimate the deformation if the bottom surface is fixed).

Hence, it is almost impossible to find a good position and extract d_{33} contribution from the complicated vertical displacement.

Fig. R5 | Top surface displacement and crystal deformation because of converse piezoelectric effect (under bipolar electric field). **a**, Crystal deformation (left) and corresponding position dependent top surface displacement (right) with free boundary condition. Here X is (100) direction. Lines with different colors in right side figures indicating d_{33} , d_{35} contribution and their combined effect. The top surface displacement of $x = 0.25$ and (b, experiment result). and $\text{LiH}_3(\text{SeO}_3)_2$ crystals (c, Simulation result, with the same space group Cc to our OIHF). **d**, Crystal deformation with bottom surface fixed boundary condition

Finally, PFM may not be necessary in this study since our existing piezoelectric measurements are confirmed reliable in several aspects. (1) The d_{33} obtained by direct and converse piezoelectric

effect show good consistence (both are around 340 pC/N under 1 Hz outer stimuli). (2) The top surface displacement diagram shows consistent features with the simulation result of crystal with the same space group (Fig. R5b and R5c). This includes the valley shape, zero displacement line and the deviation between the symmetric (where $X = 0$) and zero displacement lines (see Supplementary Text 2 for more details). This demonstrates the excellent reliability of the converse piezoelectricity measurement. (3) We have measured the d_{33} of stand PZT and PMN-Pt sample (Fig. R6) and the results (440 pc/N for PZT and 1033 pc/N for PMN-Pt) are consistent to previous reports^{3,4}. Hence, we think it is unnecessary to further conduct PFM measurements as extra evidence (especially that is regarded unreliable method for bulk crystals).

Fig. R6 | Generated current density of $x = 0.25$ crystal, PZT ceramics and PMN-0.3Pt crystals under 2MPa 1 Hz AC stress. d_{33} can be obtained by $d_{33} = J_0/2\pi fT_0$, where J_0 , f and T_0 are the magnitude of current waveform, frequency, and magnitude of stress, respectively.

Action taken: We have added Fig. R6 in supporting information (Fig. S6a).

3. The authors have shown that the material exhibits very high output voltages and currents with very high FOM values. Considering that the measurements are done on the single crystal embedded between the VHB tapes, the performance reported is for the area of the VHB tape under study than that of

single-crystal material. Hence, providing the wt% and vol% of the OIHF in the employed tape area is better.

Response: It is a good point. The wt% and vol% of the OIHF within the VHB tape area are 12.1% and 4.6%, respectively. We also would like to mention that the voltage, current and power output data shown in Fig.6 and Fig. S12 are obtained on single crystal sample without VHB tape.

Action taken: We have added the wt% and vol% in the experimental part.

4. Owing to high output voltages and current characteristics of the proof-of-concept device based on the single crystals, a detailed analysis of the piezoelectric energy harvester studies is required.

(i) To establish the piezoelectric origin of the obtained output voltages, the measurements should be performed by reversing the electrical connections to the device.

Response: We thank the reviewer's nice recommendation and have performed the harvesting measurements by reversing the electrical connections. As can be seen in Fig. R7 (added to Fig. S14e), the voltage signal is also completely reversed upon reversing the contacts. This indicates that the signal is generated from the device but not other artifices.

Fig. R7 | Voltage output with normal and reversed electrical connections on $x = 0.25$ crystal.

Action taken: We have added Fig. R7 and related discussion in supporting information. See Fig. S14e and Supplementary Text 3.

(ii) To rule out any possibility of the triboelectric effects in the measurements, single electrode or similar triboelectric-based measurements should be performed.

Response: Indeed, triboelectricity is everywhere. We have meticulously performed the measurements to eliminate the contribution from triboelectric effect. We purposely avoided using polymer in piezoelectric energy harvesting measurements on single crystal sample. Under this condition, the only possible triboelectric signal shall come from the scotch tape (see Fig R8a), which is employed to prevent direct contact between the top electrode and the metal rod of mechanical shaker. In order to evaluate its triboelectric contribution, single electrode triboelectric measurement was conducted on scotch tape by directly applying the mechanical force on the scotch tape (Fig. R8b). Results in Fig. R8c and d show that the voltage from the tape is three orders of magnitude lower compared with the piezoelectric single of crystal. We have included these measurements in our revised supporting information (see Fig. S14).

Following reviewer's suggestion, next we conducted single electrode triboelectric measurements on $x = 0.25$ crystal (see Fig. R8e). As indicated in Fig. 8f, the voltage output is even higher than the piezoelectric device in Fig. R8a. This shall be attributed to the synergistic effect between the triboelectric surface charge and the piezoelectricity induced charge, which has been carefully investigated in our previous works¹⁵. Here we did not intend to conduct more systematic triboelectric measurements of our device as it is beyond the focus of this work.

Fig. R8 | Triboelectricity relevant measurements. Experiment setups for piezoelectric and triboelectric energy harvesting of $x = 0.25$ crystal (a) and scotch tape (b). (c) Comparison between the voltage output of $x = 0.25$ crystal (piezoelectric effect) and scotch tape (triboelectric effect). (d) Zoom-in view of the triboelectric signal from c. (e) Voltage output with normal and reversed electrical connection of $x = 0.25$ crystal.

Action taken: We have added Fig. R 8a-d and related discussion in our supporting information. See Fig. S14a-d and supplementary text 3.

(iii) The energy harvesting efficiency of the device(s) must be reported.

Response: Following the reviewer's suggestion, we have calculated the efficiency of our $x = 0.25$ crystal and added details in Supplementary text 3. From previous literature, there has been no unified calculation process for piezoelectric energy harvester; and the efficiency shows dependence on various factors including device dimension and working conditions¹⁶. The reported device efficiency based on a certain piezoelectrics can vary from below 1% to 90% (see table R1). Consequently, as suggested by other high profile review paper¹⁷, the conversion efficiency is not a comprehensive figure of merit for assessing harvester performance like power output. So herein, we calculated the efficiency but not intend to use it for comparison with other piezoelectric systems.

The energy conversion efficiency (δ) of $x = 0.25$ crystal can be obtained by:

$$\delta = \frac{E_{output}}{E_{input}} = \frac{\int_0^{t_1} I^2 R dt}{\int_0^{l_1} F dl} \times 100\%$$

where I , R , t_1 , F , l , and l_1 are output current, loading resistance (1 M Ω where our crystal gets the highest power output), current pulse width, mechanical force, force exertion distance (deformation of the crystal in d_{33} direction) and the largest force exertion distance, respectively. Here l_1 can be estimated by

$$l_1 = S \times H = \frac{T}{c_{33}} \times H$$

where S , H , T , c_{33} are force induced strain, crystal thickness, stress and Young's modulus, respectively. According to above formulas and the resistance of 1 M Ω , stress of 3 MPa, c_{33} of 800 MPa, etc., the obtain efficiency is 4.06%.

Table R1| Reported efficiency values of various piezoelectric energy harvesters¹⁶.

Reference	Efficiency	Note
[23]	~50%–90%	PZT; bimorph cantilever; vibration; theoretical estimation
[24]	>80%	PZT; tube; flow-induced vibration; theoretical estimation
[28]	2.56%	PVDF film; rainbow bimorph; theoretical estimation
[29]	21.8%	PVDF nanofiber; direct deformation; experimental data
[30]	7.5%	PZT; flextensional structure; direct deformation; experimental data
[31]	>80%	PZT; fixed–fixed bimorph plate; theoretical estimation
[32]	~7%	PZT; cantilever; vibration; experimental data
[33]	0.72%	PZT; cantilever; fluid flows experimental data
[4]	5.4%/14.9%/27.5%	PZT/PMN-PT/PZN-PT; cantilever; vibration; experimental
[34]	<44%	PZT; bimorph cantilever; vibration; theoretical estimation
[35]	3.1%	PZT sandwiched between two Terfenol-D discs
[36]	1.2%	PZT; Impact-type using a rotational flywheel; experimental data
[26]	26%/<2%	PZT cantilever beam; vibration; on-resonance/off-resonance; experimental data
[37]	12.47%	PVDF/AIO-rGO beam; direct deformation; experimental data
[38]	80.3%/35.1%/15.4%	Stack/Membrane/Cantilever; ball drop impact; experimental data
[39]	10%	PZT fixed–fixed beam; ball drop impact; experimental data
[40]	5%–18%	Piezoelectric nanowires: direct deformation; experimental data

Action taken: We have added related discussion in Supplementary text 3.

(iv) A comparison of this single crystal device to its device based on bulk material (or its composite) must be performed to provide its practical utility.

Response: Unfortunately, large-sized crystals are very difficult to obtain due to impractical long time of synthesis. We note that the research of OIHF is still in early stage and focuses on discovering new structures and investigating physical properties. Indeed, single crystals are always preferred for fundamental research in intrinsic property. However, different from commercialized PVDF and PZT, currently OIHF is still new and it is challenging to achieve large-scale synthesis in practical time scale while maintaining high crystallinity. For examples, in other OIHF works, the obtained 1D crystals have very limited sizes with diameter only around 1–3 mm (Fig. R9a and b)^{18,19}, which is difficult to apply mechanical force for d_{33} -based harvesting measurements. As shown in Fig. R9c, our $x = 0.25$ crystal also has a 1D morphology with small cross section after 3 weeks natural growth (without growth in confinement condition). That is why we employed a confinement growth strategy in this study to synthesize crystal wafers with relatively larger cross section in order to achieve piezoelectric harvesting function. Hence, we would highly appreciate reviewer’s understanding about this technical challenge and current status of OIHF research.

Fig. R9 | (a,b) Photos of the (3-pyrrolinium)-MnCl₃ (a) and (pyrrolidinium)MnCl₃ (b) crystals from previous reports^{18,19} (the scale bar is 3 mm in a and 5 mm in b). (c) Photo of a x=0.25 OIHF crystal after 3 weeks synthesis.

In order to further demonstrate the utility of this material in functional devices, we fabricated a multipixel based device by embedding nine small x = 0.25 wafer crystals (size = 1 x 1 mm²) in VHB tape as multipixel based pressure sensor. Here each crystal functions as a sensing unit (pixel). The voltage response of the nine crystals were simultaneously collected when the device was tapped by finger (Fig. R10a). The exerted force on each crystal can be evaluated by taking the force-voltage graph (Fig. 10b) as a frame of reference. The resultant force is represented in a 2D force distribution Fig. R10c. Obviously, the shape and gradient of the obtained force mapping show good consistence to the real case. This offers an example for the application of this OIHF as a pressure sensor. The detailed characterization of the sensor device is out of the scope of this work and will be explored in future paper. The present paper focuses on improving the softness and intrinsic piezoelectricity of the hybrid material via molecular engineering.

Fig. R10 | **a**, Photo of the pressure sensor. **b**, Voltage-force diagram of single crystal. **c**, 2D force map of device upon tapping by finger.

5. Several tables in the manuscript and the supporting information are insufficient. (i) Fig3b and Fig4e do not include sufficient data for the best performing OIH ferro- and piezoelectric materials reported earlier.

(ii) Table S2 shows no data of g_{33} and d_{33} for OIH and their composites. There are several recent articles and reviews on these materials have appeared in the recent past.

(iii) The authors have not mentioned the TMCM cation and their family of hybrids in any of their tables, as they exhibit the highest d_{33} values for any OIH material published so far.

Response: Following reviewer's suggestion, we have revised relevant figures and table to include some recent published papers. Specific response is as follows.

Point (i), we have added the Young's modulus of PMN-Pt (see Fig. 4b), which is the materials generally regarded as inorganic piezoelectrics with the highest d_{33} . Currently there is no experiment data on the elastic modulus of $(\text{TMFM})_x(\text{TMCM})_{1-x}\text{CdCl}_3$. The elastic modulus of general organic-inorganic hybrids system are 5-70 GPa²⁰. So here we use MAPbI_3 , possibly the most well-known hybrid structure, as the example to show the Young's modulus of hybrid system (see Fig. 4b).

Point (ii), we have added more data of OIHFs to Fig. 6d. To the best of our knowledge, currently there is no piezoelectric energy harvesting study about $(\text{TMFM})_x(\text{TMCM})_{1-x}\text{CdCl}_3$. The very recent review paper in 2022 talks about energy harvesting based on hybrid materials²¹. Our purpose of Fig 6d is to provide materials' intrinsic piezoelectric performance as a guide to materials exploration, rather than comparing device-level performance. So here we have not included some of best-performance devices as many of them have special and different device structure design (for example, 3D electrodes²²) to realize higher performance. In addition, we have not included data on some hybrid perovskites (such as MAPbX_3) because the existence of their piezoelectricity is still highly debatable^{23,24}. The energy harvesting performance of some of these devices may not originate from pure piezoelectric effect. Lastly, the energy harvesting performance of these arguable piezoelectric device (power density varies from 0.2 to 36 $\mu\text{W}/\text{cm}^2$)²¹ is obviously inferior to ours (1100 $\mu\text{W}/\text{cm}^2$).

Point (iii): Following reviewer's suggestion, we have mentioned the d_{33} value of $(\text{TMFM})_x(\text{TMCM})_{1-x}\text{CdCl}_3$ in the revised introduction and added more information about OIHFs in Table S2. We also emphasized in Fig 3b caption the high d_{33} values of PMN-Pt (up to 2000 pC/N) and $(\text{TMFM})_x(\text{TMCM})_{1-x}\text{CdCl}_3$ (1540 pC/N). However, we decided not to add the data into Fig. 3b because

of two reasons: i) Here our focus is to demonstrate that molecular engendering is effective in making a hybrid system possess both large d_{33} that is comparable to the benchmark of inorganic piezoelectric (PZT) and excellent mechanical softness similar to the piezoelectric polymer PVDF, rather than to claim the high d_{33} so to compare it with the highest performance piezoelectrics. ii) After adding the data of PMN-Pt and $(\text{TMFM})_x(\text{TMCM})_{1-x}\text{CdCl}_3$, all the data points in Fig. 3b will be squeezed to a small region ($d_{33} < 500$) and lose the clarity. We highly respect reviewer's comment and hope for understanding.

Action taken: We have made the following changes:

- 1) Added piezoelectricity information about OIHF including $(\text{TMFM})_{0.26}(\text{TMCM})_{0.74}\text{CdCl}_3$ in Supplementary Table S2;
- 2) Added statement "For instance, the MPB induced large $d_{33} = 1540$ pC/N of $(\text{TMFM})_{0.26}(\text{TMCM})_{0.74}\text{CdCl}_3$ " in Introduction;
- 3) Added statement of "For clarity, data for PMN-Pt (d_{33} up to 2000 pC/N) and $(\text{TMFM})_{0.26}(\text{TMCM})_{0.74}\text{CdCl}_3$ ($d_{33} = 1540$ pC/N), which are generally regarded as the piezoelectrics with the highest d_{33} in inorganic and organic system, respectively, is not included in b" in the caption of Fig. 3b.
- 4) Added PMN-Pt data into Fig. 4b and added more OIHF systems into Fig. 6d.

6. Though the obtained output characteristics (voltage, power density) of the device are high, the voltages stored in the capacitors are very low and, in fact much lower than those obtained from several hybrid bulk composite materials (with no directional behaviour). Hence, the authors must provide the data from the bulk sample and compare them with the literature known examples derived from hybrid materials.

Response: The stored energy of our single crystal device should be in principle lower than soft polymer contained device system. We argue that this phenomenon is reasonable because the energy storage performance depends not only on the piezoelectricity but also on how the device is fabricated (see more elaboration below). So, it may not be fair to use different device systems to compare material's piezoelectricity. For many hybrid ferroelectrics related devices, the OIHF's are embedded in soft polymer. This will largely increase the mechanical softness of the device and consequently the total input energy, leading to apparently high energy storage performance. Let's take the previous $(\text{Ph}_3\text{MeP})_4\text{Ni}(\text{NCS})_6$ device work for example²⁵, where the material piezoelectricity, device

mechanical property and device energy harvesting performance were relatively systematically studied. The power density of that polymer-(Ph_3MeP) $_4\text{Ni}(\text{NCS})_6$ device is $50.26 \mu\text{W}/\text{cm}^2$, which is quite high in all OIHF based systems (0.2 to $75 \mu\text{W}/\text{cm}^2$)²¹. HOWEVER, in this work, the piezoelectrics was embedded in TPU polymer, and the Young's modulus of the device is largely decreased. As shown in Fig. R11, at 3 MPa stress, the produced strain reaches around 25% while for our crystal the strain is: stress/Young's modulus = $3\text{MPa}/800\text{MPa} = 0.375\%$. This implies that input mechanical energy (= force \times sample dimension \times produced strain) can be two orders of magnitudes higher than our device with similar device dimension. Given the truth that the stored energy is proportional to input energy, that results in significantly higher stored energy than ours. However, the d_{33} of (Ph_3MeP) $_4(\text{Ni}(\text{NCS})_6)$ is very low, around $8 \text{ pC}/\text{N}$. This indicates the storage property of piezoelectric composites is not directly related to the intrinsic piezoelectric energy harvesting property of the embedded ferroelectrics.

We hope for the reviewer's kind understanding that the focus of our work is to propose a molecular engineering method to effectively improve the piezoelectricity and mechanical softness of hybrids system, which is similar to these two material-science oriented OIHF's works published in *Matter* and *JACS*^{26,27}. We fully agree that composites nano generator (in question 4 (iv)) and energy storage device (this question) are all important applications of piezoelectrics. To push our device performance in specific aspect, our ferroelectric wafer crystals can also be embedded in certain polymers, which will be investigated in future device-oriented study.

Fig. R11 | Strain-stress curve of (Ph_3MeP) $_4\text{Ni}(\text{NCS})_6$ -TPU system, from reference²⁵.

In addition, in our initial submission, the charging was conducted by low forces. Now we have conducted the measurement at higher stress around 3 MPa. As shown in Fig. R12a, the capacitor voltage of our crystal (0.68 V for $22 \mu\text{F}$) is comparable to that of (Ph_3MeP) $_4(\text{Ni}(\text{NCS})_6)$ -TPU system

(Fig. R12b, around 0.5-1 V at 4s) even with much lower input energy. Currently there are relatively few studies about the intrinsic piezoelectric harvesting property on crystal-form OIHFs. We found only three reports²⁶⁻²⁸ which have voltage output in the range of 2-5V (see two of them in Fig. R12c and d), much lower than that of our crystal (210 V). These comparisons reveal the excellent intrinsic piezoelectric energy harvesting property of our solid solution.

Fig. R12 | **a** and **b**, Traces of capacitor charging of $x = 0.25$ crystal (**a**) and $(\text{Ph}_3\text{MeP})_4(\text{Ni}(\text{NCS})_6)\text{-TPU}$ composites (**b**)²⁵. The orange circle marks the possible range of the voltage with 22 μF capacitor at 4s. **c** and **d**, Voltage output of $(\text{FMTMA})\text{PbCl}_2\text{I}_2$ ²⁶ and $(\text{TMFM})\text{FeBr}_4$ crystal²⁷.

Action taken: We have added Fig. R12a to supporting information (Fig. S12e).

7. The stored energies and measured charges in the capacitors should also be provided.

Response: Following the reviewer's suggestion, we have done energy storage measurement as shown in Fig. R12a. The stored energies and measured charge can be calculated by:

$$E_n = \frac{1}{2} C \times V^2$$

$$Q = C \times V$$

The calculated energy and charge are 5.39 μJ and 14.2 μC , respectively.

Action taken: We have added the above information into Supplementary Text 3.

8. Most of the Figures in the manuscript are cluttered with multiple images and long figure captions. They must be split. Also, figure captions should be elaborated a bit more as this reviewer could not make a smooth correlation of the figure captions with the related text in the main manuscript.
Response: We sincerely apologize for the inconvenience. Following reviewer's suggestion, we have split Fig 2 and 6 into four figures and revised captions accordingly.

Reviewer #3 (Remarks to the Author):

In this manuscript, the authors incorporated iodide into an organic-inorganic hybrid ferroelectric $C_6H_5N(CH_3)_3CdBr_2Cl$, which weakens the metal-halide bonds. The obtained $C_6H_5N(CH_3)_3CdBr_2Cl_{1-x}I_x$ ferroelectric solid solution shows outstanding piezoelectric and energy harvesting performances with g_{33} and output power density up to 3595×10^{-3} Vm/N and 11 W/m², respectively, much higher than those of widely used piezoelectric PZT. This work sheds light on the exploration of high-performance soft piezoelectrics. I recommend publishing it on Nature Communications after addressing the following issues.

Response: We thank the reviewer for his/her positive comments.

1. In the introduction part, the authors only discussed the g_{33} of inorganic PZT and organic PVDF. The works on the g_{33} of organic-inorganic hybrid ferroelectrics were not mentioned (such as J. Am. Chem. Soc. 2020, 142, 1077–1082).

Response and action taken: Thank you for this reminder. We have cited now the JACS paper in our revised manuscript (see ref. 22) to offer readers a more comprehensive research background of hybrid piezoelectrics.

2. The piezoelectric performance of ferroelectric solid solution $C_6H_5N(CH_3)_3CdBr_3xCl_{3(1-x)}$ was previously reported by the authors. They should mention this in the introduction part as well.

Response and action taken: Follow reviewer's suggestion, we have mentioned this in our revised manuscript in Introduction part:

“For instance, the MPB induces the large $d_{33} = 1540$ pC/N of $(TMFM)_{0.26}(TMCM)_{0.74}CdCl_3$ ²³ and the large shear strain/piezoelectricity of $C_6H_5N(CH_3)_3CdBr_xCl_{3-x}$ solid solution²⁴.”

3. For the d_{33} , the authors are suggested to measure it by the Berlincourt method.

Response: Excellent comment. Referee #1 asked exactly the same question (No. 3). Now we have added details about the Berlincourt piezometer method and related equations to Supplementary

Text 1 and data to a new Fig. S6a. To cut short this letter, you may want to refer to our answer on page 2. We also summarize below.

We are sorry for not explaining the measurement in detail in the Method part. Our system is a home-made Berlincourt piezometer, which is in principle identical to the original Berlincourt piezometer but with tuneable stress frequency and amplitude (see Fig R1a). The system provides high accuracy and has enabled the successful characterisation of artificially induced piezoelectricity in Schottky junctions (see Nature 584, 377–381, 2020)¹. Under the condition of constant outer electric field (which is zero), the d_{33} of the tested sample are obtained from:

$$d_{33} = J_0/2\pi fT_0$$

In addition, we also measured the piezoelectric d_{33} values of PZT ceramic and PMN-0.3Pt crystal (Fig. R1b). The obtained d_{33} of 440 pC/N for PZT and 1033 pC/N for PMN-0.3PT are consistent with previous reports^{3,4}, indicating the good reliability of our measuring system.

Action taken: We have added a new Fig. S6 and added PZT and PMN-Pt data. Detail explanation was added as Supplementary Text 1.

4. The Young's modulus of organic-inorganic $C_6H_5N(CH_3)_3CdBr_2Cl_{0.75}I_{0.25}$ is even lower than that of pure organic PVDF. The authors should double check the Young's modulus of $C_6H_5N(CH_3)_3CdBr_2Cl_{0.75}I_{0.25}$. The theoretical value is suggested to be provided.

Response: Following reviewer's suggestion, we have recalibrated our nanoindentation measurement setup using a standard Pt sample (Fig. R14), and the obtained Young's modulus of 155 GPa is consistent with previous reports^{31,32}. Furthermore, we measured three $x = 0.25$ crystals from different growth runs. The obtained values of 0.72-0.9 GPa are close and similar to previous results. Yes, the Young's modulus of our $x = 0.25$ crystals is obviously lower than PVDF. As discussed in text, the softness of this OIHF shall originate from the composition flexibility (on halide) and its induced effect, which can continuously weaken the Cd-X1 and other metal-halide bonds until the phase boundary (around $x = 0.25$ composition). Such molecular engineering and bond weakening effect is not found in general PVDF molecular chain. This is the main reason behind the superior softness of this hybrid material compared to PVDF.

Fig. R14 | Strain-stress curve of standard Pt metal (a) and $x = 0.25$ crystals (b).

Next, we simulated the theoretical Young's moduli for both our $C_6H_5N(CH_3)_3CdBr_2Cl_{0.75}I_{0.25}$ crystal and PVDF. As shown in Table R2 and R3, the simulated c_{33} value for $x = 0.25$ crystal is 18.5 GPa and 62.9 GPa for PVDF. On one hand, the trend is consistent with our experiments, that our hybrid solid solution is softer than PVDF. It corroborates the reliability of our nanoindentation measurement. On the other hand, the simulated values are both obviously larger than the experiment result. This has been a common observation, as demonstrated in various similar large molecule systems³³⁻³⁵. For example, the simulated Young's modules of PVDF are around tens to few hundreds GPa from previous reports^{33,36-38}, which is much larger than its experimental value (2 GPa)³⁹.

Table R2 | Simulated elastic modulus matrix of $C_6H_5N(CH_3)_3CdBr_2Cl_{0.75}I_{0.25}$ (unit: GPa). c_{33} is marked by red.

Direction	1	2	3	4	5	6
1	22.63253	8.32768	8.85910	-0.04053	0.1453	-0.70952
2	8.32768	24.88585	6.82322	0.21560	0.41519	-0.06503

3	8.85910	6.82322	18.49419	0.33627	0.10296	-0.18136
4	-0.04053	0.21560	0.33627	2.20557	0.00563	-0.45451
5	0.1453	0.41519	0.10296	0.00563	4.96580	0.04429
6	-0.70952	-0.06503	-0.18136	-0.45451	0.04429	4.34158

Table R3 | Simulated elastic modulus matrix of PVDF (unit: GPa). c_{33} is marked by red.

Direction	1	2	3	4	5	6
1	348.21200	2.93705	10.84387	-	-	-
2	2.93705	58.19075	11.26782	-	-	-
3	10.84387	11.26782	62.93011	-	-	-
4	-	-	-	7.41583	-	-
5	-	-	-	-	14.31248	-
6	-	-	-	-	-	16.48022

Action taken: We have added Fig. R14 in supporting information as Fig. S8 b-c.

5. In figure S4, the change of dielectric constant around T_c is very small, which is not common for ferroelectric crystals. Please discuss the reason.

Response: We thank the reviewer for his great care in this intrinsic phenomenon. The phase change related to the dielectric anomaly in Fig. S4 (Fig. R15a here) is not the Curie temperature for ferroelectric-paraelectric phase transition. Both low and high temperature phases are ferroelectric belonging to the space groups Cc and $Ama2$. Consequently, the dielectric change induced by this transition shall not have features similar to a ferroelectric-paraelectric phase transition at Curie temperature. Similar type of ferroelectric-ferroelectric phase transition happens in oxides such as

BTO and PIN-PMN-Pt (see the step-like behaviour around 130 °C of Fig. R15b), in which the dielectric constant changes are very small near the transition temperature^{40,41}. Hence, the dielectric behaviour in our case in Fig. S4 should be reasonable. We have added relevant discussion in the caption of Fig. S4.

Fig. R15 | **a**, Dielectric behavior of $x = 0.25$ crystal as a function of temperature. **b**, Temperature dependent dielectric permittivity and loss of PIN-PMN-Pt⁴⁰.

Action taken: We have added a short explanation in the caption of Fig. S4 as follows:

“This dielectric behavior along the (001) direction, i.e., small change in dielectric constant near to transition temperature, is similar to those of classical ferroelectric systems (such as BTO and PIN-PMN-Pt) during the ferroelectric-ferroelectric phase transition^{17,18}.”

Reference

- 1 Yang, M.-M. *et al.* Piezoelectric and pyroelectric effects induced by interface polar symmetry. *Nature* **584**, 377-381, doi:10.1038/s41586-020-2602-4 (2020).
- 2 Damjanovic, D. & Demartin, M. The Rayleigh law in piezoelectric ceramics. *J. Phys. D: Appl. Phys.* **29**, 2057-2060, doi:10.1088/0022-3727/29/7/046 (1996).
- 3 Li, F., Xu, Z., Wei, X. & Yao, X. Determination of temperature dependence of piezoelectric coefficients matrix of lead zirconate titanate ceramics by quasi-static and resonance method. *J. Phys. D: Appl. Phys.* **42**, 095417, doi:10.1088/0022-3727/42/9/095417 (2009).
- 4 Shrout, T. R., Chang, Z. P., Kim, N. & Markgraf, S. Dielectric behavior of single crystals near the (1-X) Pb(Mg_{1/3}Nb_{2/3})O₃-(x) PbTiO₃ morphotropic phase boundary. *Ferroelectrics Letters Section* **12**, 63-69, doi:10.1080/07315179008201118 (1990).
- 5 Liao, W.-Q. *et al.* A molecular perovskite solid solution with piezoelectricity stronger than lead zirconate titanate. *Science* **363**, 1206-1210, doi:10.1126/science.aav3057 (2019).
- 6 You, Y. M., Liao, W. Q., Zhao, D., Ye, H. Y., Zhang, Y., Zhou, Q. An organic-inorganic perovskite ferroelectric with large piezoelectric response. *Science* **357**, 306-309 (2017).
- 7 Alexe, M. & Hesse, D. Tip-enhanced photovoltaic effects in bismuth ferrite. *Nat. Commun.* **2**, 256, doi:10.1038/ncomms1261 (2011).
- 8 Gruverman, A., Alexe, M. & Meier, D. Piezoresponse force microscopy and nanoferroic phenomena. *Nat. Commun.* **10**, 1661, doi:10.1038/s41467-019-09650-8 (2019).
- 9 Biegalski, M. D. *et al.* Strong strain dependence of ferroelectric coercivity in a BiFeO₃ film. *Appl. Phys. Lett.* **98**, 142902, doi:10.1063/1.3569137 (2011).
- 10 Aizu, K. Possible Species of "Ferroelastic" Crystals and of Simultaneously Ferroelectric and Ferroelastic Crystals. *J. Phys. Soc. Jpn.* **27**, 387-396, doi:10.1143/jpsj.27.387 (1969).
- 11 Hu, Y. *et al.* Ferroelastic-switching-driven large shear strain and piezoelectricity in a hybrid ferroelectric. *Nature Mater.* **20**, 612-617, doi:10.1038/s41563-020-00875-3 (2021).
- 12 Damjanovic, D. Stress and frequency dependence of the direct piezoelectric effect in ferroelectric ceramics. *J. Appl. Phys.* **82**, 1788-1797, doi:10.1063/1.365981 (1997).
- 13 Abdollahi, A., Domingo, N., Arias, I. & Catalan, G. Converse flexoelectricity yields large piezoresponse force microscopy signals in non-piezoelectric materials. *Nat. Commun.* **10**, 1266, doi:10.1038/s41467-019-09266-y (2019).
- 14 IEEE Standard on Piezoelectricity. *ANSI/IEEE Std 176-1987, 0_1*, doi:10.1109/IEEESTD.1988.79638 (1988).
- 15 Parida, K., Bhavanasi, V., Kumar, V., Bendi, R. & Lee, P. S. Self-powered pressure sensor for ultra-wide range pressure detection. *Nano Research* **10**, 3557-3570, doi:10.1007/s12274-017-1567-6 (2017).
- 16 Yang, Z., Erturk, A. & Zu, J. On the efficiency of piezoelectric energy harvesters. *Extreme Mechanics Letters* **15**, 26-37, doi:<https://doi.org/10.1016/j.eml.2017.05.002> (2017).
- 17 Yang, Z., Zhou, S., Zu, J. & Inman, D. High-Performance Piezoelectric Energy Harvesters and Their Applications. *Joule* **2**, 642-697, doi:<https://doi.org/10.1016/j.joule.2018.03.011> (2018).
- 18 Zhang, Y. *et al.* Highly Efficient Red-Light Emission in An Organic-Inorganic Hybrid Ferroelectric: (Pyrrolidinium)MnCl₃. *J. Am. Chem. Soc.* **137**, 4928-4931, doi:10.1021/jacs.5b01680 (2015).
- 19 Ye, H. Y. *et al.* High-Temperature Ferroelectricity and Photoluminescence in a Hybrid Organic-Inorganic Compound: (3-Pyrrolinium)MnCl₃. *J. Am. Chem. Soc.* **137**, 13148-13154, doi:10.1021/jacs.5b08290 (2015).

- 20 Ji, L.-J., Sun, S.-J., Qin, Y., Li, K. & Li, W. Mechanical properties of hybrid organic-inorganic perovskites. *Coord. Chem. Rev.* **391**, 15-29, doi:<https://doi.org/10.1016/j.ccr.2019.03.020> (2019).
- 21 Vijayakanth, T., Liptrot, D. J., Gazit, E., Boomishankar, R. & Bowen, C. R. Recent Advances in Organic and Organic–Inorganic Hybrid Materials for Piezoelectric Mechanical Energy Harvesting. *Adv. Funct. Mater.* **32**, 2109492, doi:<https://doi.org/10.1002/adfm.202109492> (2022).
- 22 Gu, L. *et al.* Enhancing the current density of a piezoelectric nanogenerator using a three-dimensional intercalation electrode. *Nat. Commun.* **11**, 1030, doi:10.1038/s41467-020-14846-4 (2020).
- 23 Liu, Y. *et al.* Chemical nature of ferroelastic twin domains in CH₃NH₃PbI₃ perovskite. *Nature Mater.* **17**, 1013-1019, doi:10.1038/s41563-018-0152-z (2018).
- 24 Huang, J., Yuan, Y., Shao, Y. & Yan, Y. Understanding the physical properties of hybrid perovskites for photovoltaic applications. *Nature Reviews Materials* **2**, 17042, doi:10.1038/natrevmats.2017.42 (2017).
- 25 Vijayakanth, T., Ram, F., Praveenkumar, B., Shanmuganathan, K. & Boomishankar, R. Piezoelectric Energy Harvesting from a Ferroelectric Hybrid Salt [Ph₃MeP]₄[Ni(NCS)₆] Embedded in a Polymer Matrix. *Angew. Chem. Int. Ed.* **n/a**, doi:10.1002/anie.202001250.
- 26 Zhang, Z.-X. *et al.* Organometallic-Based Hybrid Perovskite Piezoelectrics with a Narrow Band Gap. *J. Am. Chem. Soc.* **142**, 17787-17794, doi:10.1021/jacs.0c09288 (2020).
- 27 Zhang, Y., Song, X.-J., Zhang, Z.-X., Fu, D.-W. & Xiong, R.-G. Piezoelectric Energy Harvesting Based on Multiaxial Ferroelectrics by Precise Molecular Design. *Matter* **2**, 697-710, doi:<https://doi.org/10.1016/j.matt.2019.12.008> (2020).
- 28 Shi, C. *et al.* Large Piezoelectric Response in Hybrid Rare-Earth Double Perovskite Relaxor Ferroelectrics. *J. Am. Chem. Soc.* **142**, 9634-9641, doi:10.1021/jacs.0c00480 (2020).
- 29 Berlincourt, D. & Krueger, H. H. A. Domain Processes in Lead Titanate Zirconate and Barium Titanate Ceramics. *J. Appl. Phys.* **30**, 1804-1810, doi:10.1063/1.1735059 (1959).
- 30 Stewart, M. & Cain, M. (Springer Dordrecht, 2014).
- 31 Merker, J., Lupton, D., Topfer, M. & Knake, H. High temperature mechanical properties of the platinum group metals. *Platinum Metals Review(UK)* **45**, 74-82 (2001).
- 32 Salvadori, M. C., Brown, I. G., Vaz, A. R., Melo, L. L. & Cattani, M. Measurement of the elastic modulus of nanostructured gold and platinum thin films. *Phys. Rev. B* **67**, 153404, doi:10.1103/PhysRevB.67.153404 (2003).
- 33 Sun, F.-C., Dongare, A. M., Asandei, A. D., Pamir Alpay, S. & Nakhmanson, S. Temperature dependent structural, elastic, and polar properties of ferroelectric polyvinylidene fluoride (PVDF) and trifluoroethylene (TrFE) copolymers. *J. Mater. Chem. C* **3**, 8389-8396, doi:10.1039/C5TC01224D (2015).
- 34 Lin, T., Liu, X.-Y. & He, C. Ab Initio Elasticity of Poly(lactic acid) Crystals. *The Journal of Physical Chemistry B* **114**, 3133-3139, doi:10.1021/jp911198p (2010).
- 35 Setoodeh, A. R. & Farahmand, H. Continuum-DFT multiscale model to investigate linear/nonlinear anisotropic mechanical characterization of crystal phase of nylon-6, 6. *Mech. Mater.* **117**, 181-191, doi:<https://doi.org/10.1016/j.mechmat.2017.11.010> (2018).
- 36 Pei, Y. & Zeng, X. C. Elastic properties of poly(vinylidene fluoride) (PVDF) crystals: A density functional theory study. *J. Appl. Phys.* **109**, 093514, doi:10.1063/1.3574653 (2011).
- 37 Omote, K., Ohigashi, H. & Koga, K. Temperature dependence of elastic, dielectric, and piezoelectric properties of “single crystalline” films of vinylidene fluoride trifluoroethylene copolymer. *J. Appl. Phys.* **81**, 2760-2769, doi:10.1063/1.364300 (1997).

- 38 Carbeck, J. D. & Rutledge, G. C. Temperature dependent elastic, piezoelectric and pyroelectric properties of β -poly(vinylidene fluoride) from molecular simulation. *Polymer* **37**, 5089-5097, doi:[https://doi.org/10.1016/0032-3861\(96\)00366-7](https://doi.org/10.1016/0032-3861(96)00366-7) (1996).
- 39 Sengupta, D. *et al.* Characterization of single polyvinylidene fluoride (PVDF) nanofiber for flow sensing applications. *AIP Advances* **7**, 105205, doi:10.1063/1.4994968 (2017).
- 40 Zhang, S. *et al.* Electromechanical characterization of $\text{Pb}(\text{In}_{0.5}\text{Nb}_{0.5})\text{O}_3\text{-Pb}(\text{Mg}_{1/3}\text{Nb}_{2/3})\text{O}_3\text{-PbTiO}_3$ crystals as a function of crystallographic orientation and temperature. *J. Appl. Phys.* **105**, 104506, doi:10.1063/1.3131622 (2009).
- 41 Merz, W. J. The Electric and Optical Behavior of BaTiO_3 Single-Domain Crystals. *Phys. Rev.* **76**, 1221-1225, doi:10.1103/PhysRev.76.1221 (1949).

REVIEWERS' COMMENTS

Reviewer #1 (Remarks to the Author):

Good response to referee comments and paper is much improved. Happy to accept.

Reviewer #2 (Remarks to the Author):

The authors have comprehensively addressed the concerns and comments of all the reviewers. The only concern is that the authors have not attempted to perform PFM and show the results. While this reviewer understands that the PFM results on the bulk crystals can not give reliable data, the motivation was to encourage the authors to prepare the thin crystallites of their sample on a Pt or Si surface and see the domain structures. Hence, this reviewer encourages the authors to perform PFM experiments if possible.

Nevertheless, this reviewer is satisfied with the revisions and responses provided by the authors for the rest of the comments. Hence, I would be pleased to recommend the acceptance of the manuscript in the present form

Reviewer #3 (Remarks to the Author):

I am satisfied with the authors' responses to my comments. I would recommend acceptance of the current version of manuscript for publication.

Response to Reviewers' Comments

Reviewer #1 (Remarks to the Author):

Good response to referee comments and paper is much improved. Happy to accept.

Response: Thank you! We appreciate your strong support to this paper and all the useful comments.

Reviewer #2 (Remarks to the Author):

The authors have comprehensively addressed the concerns and comments of all the reviewers. The only concern is that the authors have not attempted to perform PFM and show the results. While this reviewer understands that the PFM results on the bulk crystals can not give reliable data, the motivation was to encourage the authors to prepare the thin crystallites of their sample on a Pt or Si surface and see the domain structures. Hence, this reviewer encourages the authors to perform PFM experiments if possible.

Nevertheless, this reviewer is satisfied with the revisions and responses provided by the authors for the rest of the comments. Hence, I would be pleased to recommend the acceptance of the manuscript in the present form

Response: We thank the reviewer for positive suggestion and the great patience in reviewing and our manuscript and response.

As per your comment, as summarized in our previous response letter, the non-180° ferroelastic switch (domain feature) has been confirmed by three different experimental methods. Adding a fourth proof is good but could be redundant.

Reviewer #3 (Remarks to the Author):

I am satisfied with the authors' responses to my comments. I would recommend acceptance of the current version of manuscript for publication.

Response: We thank the reviewer for his/her positive comment as well as the help in improving the manuscript.